# Scenario approach for the seasonal forecast of Kharif flows from Upper Indus Basin

Muhammad Fraz Ismail and Wolfgang Bogacki

Department of Civil Engineering, Koblenz University of Applied Sciences, Germany.

*Correspondence to*:  W. Bogacki (bogacki@hs-koblenz.de)

**Abstract.** Snow and glacial melt runoff are the major sources of water contribution from the high mountainous terrain of Indus river upstream of the Tarbela reservoir. A reliable forecast of seasonal water availability for the Kharif cropping season (April – September) can pave the way towards the better water management and subsequently boost the agro-economy of Pakistan. The use of degree-day models in conjunction with the satellite based remote sensing data for the forecasting of seasonal snow and ice melt runoff has proved to be a suitable approach for the data scarce regions. In the present research, Snowmelt Runoff Model (SRM) has not only been enhanced by incorporating the "glacier (G)" component but also applied for the forecast of seasonal water availability from the Upper Indus Basin (UIB). Excel based SRM+G takes into account of separate degree-day factors for snow and glacier melt processes. All year simulation runs with SRM+G for the period 2003 – 2014 result in an average flow component distribution of 53%, 21%, and 26% for snow, glacier and rain respectively. The UIB has been divided into Upper and Lower parts because of the different climatic conditions in the Tibetan plateau. The scenario approach for seasonal forecasting, which like Ensemble Streamflow Prediction uses historic meteorology as model forcings, has proven to be adequate for long-term water availability forecasts. The accuracy of the forecast with a MAPE of 9.5% could be slightly improved compared to two existing operational forecasts for the UIB and the bias could be reduced to -2.0%. However, the association between forecasts and observations as well as the skill in predicting extreme conditions is rather weak for all three models, which motivates further research on the selection of a subset of ensemble members according to forecasted seasonal anomalies.

# 1 Introduction

Mountains are the water towers of the world. They are the biggest resource of freshwater to half of the world's population fulfilling their needs for irrigation, industry, domestic and hydropower (Viviroli et al., 2007). The Indus River on which Pakistan's socio-economic development depends can be termed as the bread basket of Pakistan
(Clarke, 2015). Due to agrarian economy, Pakistan's agriculture share in water usage is about 97%, which is well above the global average of about 70% (Akram, 2009). In Pakistan, Indus River System Authority (IRSA) decides the provincial water shares according to the Water Apportionment Accord (WAA) of 1991 and provincial irrigation departments subsequently determine the seasonal water allocation to the different canal command areas depending upon the water availability forecast carried out at the end of March for the forthcoming Kharif cropping season
(April-September). A reliable seasonal forecast of the water availability from snow and glacial melt is therefore of utmost importance for the agricultural production and efficient water management.

But on the other hand snowmelt runoff modelling in mountainous regions faces the challenge of data scarcity as well as the uncertainty in parameter calibration (Pellicciotti et al., 2012). The need of the hour is to not only develop such a hydrological model which has the capability to cater both snow and glacial melt component but also a
reliable forecast technique which could help the water managers and policy makers for enhancing the water resources management in future. The present paper focuses on the implementation of Snowmelt Runoff Model (SRM) including the glacier melt component (+G) based on the methodology proposed by Schaper et al. (1999), which is an important value addition to the existing ExcelSRM version (Bogacki and Hashmi, 2013) of WinSRM (Martinec et al., 2011) model. In the earlier studies on Upper Indus Basin (UIB) and its sub-catchments e.g.
Immerzeel et al., (2010), Tahir et al., (2011), Butt and Bilal (2011) and Adnan et al., (2016) have used only SRM standard version while glaciers are dealt by taking them as a part of snow covered area. The underestimation of flows in periods associated with glacier melt contribution as pointed out by Tahir et al., (2011) has now been dealt with by incorporation of a glacier melt component. A unique methodology has been adopted to deal with the early fading of snow cover area from the Tibetan plateau by separating the whole UIB into two sub-catchments which is
not implemented in original WinSRM.

Ensemble Streamflow Prediction (ESP), developed at the U.S. National Weather Service (Day, 1985) is widely used to generate probabilistic long-term stream flow forecasts. As already successfully applied in the Upper Jhelum basin (Bogacki and Ismail, 2016), a scenario approach is used for seasonal flow forecasting in the UIB, which has much similarity to ESP. It also uses historical meteorology as model forcings, however like the other operational
forecast models for UIB, it is mainly focussed on a deterministic forecast of total Kharif inflow to the Tarbela reservoir.

# 2 Materials and Methods

## 2.1 Study area

The upper catchments of the Indus River basin (Figure 1) primarily feed the Tarbela reservoir, which is the larger of the only two major reservoirs in Pakistan. The Upper Indus Basin has an area of about 173,345 km$^2$ of which approx. 11.5% is covered by perennial glacial ice (Tahir et al., 2011). At the end of most winters nearly the entire

UIB above 2,200m a.s.l is covered with snow, resulting more than 60% of annual flow in the Upper Indus River is contributed by the snowmelt (Bookhagen and Burbank, 2010). The distribution of monthly inflows to the Tarbela reservoir (see Figure 2) shows that these flows tend to rise progressively as melting temperatures advance into areas of higher snowpack at the higher elevations. Indus River starts rising gradually in March reaching its maximum in July, while peak flood events usually occur during the monsoon season in July – September. When by the end of July the flows reduce due to diminished snow cover in the lower catchment, the high altitude glacierised basins become important contributors to the flows due to first melting of their seasonal snow cover and when the snow has vanished then melting of the glacier ice. According to Tahir et al. (2011) glacial melt dominates the flows of the largest tributaries of Indus River, i.e. Chitral, Gilgit, Hunza, Braldu and Shyok rivers.

## 2.2 Model Structure

The Snowmelt Runoff Model (SRM) (Martinec, 1975) is a semi-distributed, lumped temperature-index model which is specifically designed to simulate the runoff in snow-dominated catchments that has been successfully applied in more than hundred snow-driven basins around the globe (Martinec et al., 2011). Input variables of SRM are daily values of air temperature, precipitation, and snow covered area. The catchment is usually subdivided into elevation zones of about 500 m each and the input variables are distributed accordingly. The total daily amount of water produced from snowmelt and rainfall in the catchment is superimposed on the calculated recession flow according to the equation (1):

$$Q_{n+1} = \sum_{i=1}^{m} \left\{ [M_{n,i} + R_{n,i}] . \frac{10000}{86400} A_i \right\} . (1 - k_{n+1}) + Q_n k_{n+1} \qquad (1)$$

Where, $Q$ is the average daily discharge [$m^3s^{-1}$], $M$ and $R$ are the daily runoff depth originating from snowmelt and rainfall [cm $d^{-1}$], $A$ is the total area of the elevation zone [$km^2$], $k$ is the recession coefficient [-], $n$ is the index of the simulation day and $i$, $m$ are the indices and total number of elevation zones respectively. Daily runoff from snowmelt and rainfall is calculated by equations (2) and (3):

$$M_{n,i} = c_{Sn,i}. a_{Sn,i}. T_{n,i}. S_{n,i} \qquad (2)$$
$$R_{n,i} = c_{Rn,i}. P_{n,i} \qquad (3)$$

Where, $c_S$ and $c_R$ are the runoff coefficients [-] for snowmelt and rain, $a_S$ is the degree-day factor for snow [cm °C$^{-1}$ d$^{-1}$], $T$ the number of degree-days [°C d] for each elevation zone, $S$ the ratio of the snow covered area to the total area [-] and $P$ the daily precipitation [cm $d^{-1}$].

Schaper et al., (1999) introduced an enhancement in the original SRM approach by incorporating the separate glacial melt component in the model. In addition to the variables used by SRM it also considers the area covered by exposed, i.e. not snow covered, glaciers. An additional melt component is added to equation (1) that takes into account the specific degree-day factors for glaciers according to equation (4):

$$G_{n,i} = c_{Gn,i} . a_{Gn,i} . T_{n,i} . S_{Gn,i} \qquad\qquad (4)$$

Where, $G$ is the daily melt [cm d$^{-1}$] from exposed glaciers in each elevation zone, $c_G$ is the runoff coefficient [-] and $a_G$ is the degree-day factor [cm °C$^{-1}$ d$^{-1}$] for glaciers, and $S_G$ is the ratio of the exposed glacier area to the total area [-].

This model was tested in several basins and found with high accuracy even in basins with 67% glacier areas on three alpine basins Rhine-Felsberg, Rhône-Sion and Ticino-Bellinzona in Switzerland (Schaper and Seidel, 2000). Apart from the improvement of the runoff modelling, the independent computation of glacier melt is an important step towards evaluations of glacier behaviour with regard to climate change.

The glacier melt component according to equation (4) was incorporated into the existing ExcelSRM further referred to as SRM+G. This extension requires the glacier exposed area as an additional daily input variable and respective model parameters as given in equation (4).

An additional enhancement is the possibility to split the watershed into different sub-catchments. This feature is realised by adding the pre-calculated outflow of a sub-catchment obtained by a separate simulation to the discharge of the downstream sub-catchment. The travel time can be considered by applying a time-lag to the daily discharge time-series.

## 2.3 Splitting the UIB into two sub-catchments

In the Karakorum – Western Himalayas region snow accumulates during winter and reaches its maximum extension in February/March. Higher altitudes typically have a 90% – 100% snow cover that stays more or less constant until melting starts in spring. There is however a characteristic bias between the north-western part of the UIB where at altitudes above 4,000 m a.s.l. the snow covered area usually starts gradually decreasing in March, while in the south-eastern part namely the Tibetan plateau at the same altitudes the snow cover is fading away very soon. This bias leads to an inevitable under-estimation in forecasting the snowmelt dominated Early Kharif flows (see Chap. 3.1), which motivates the splitting of the UIB into two sub-catchments.

Ideally, the catchment should be split right downstream of the Tibetan plateau. However, as the first gauging station where daily flow data was available is Kharmong gauging station, when the Upper Indus River has entered into Pakistan, this location was chosen to split the UIB in an upstream and a downstream sub-catchment, namely the Lower and Upper UIB (Figure 1). The hypsometric characteristics including the number of elevation zones and their corresponding areas of both the sub-catchments are shown in Figure 3.

According to the two sub-catchments, two separate SRM+G models were created. For each simulation, first the Upper UIB model is run in order to simulate flows at Kharmong. These flows are then superimposed to the flows calculated by the Lower UIB model using a time-lag between Kharmong and Tarbela that was estimated by the Kirpich equation (5) (Kirpich, 1940; USAD, 2010)

$$t = 0.00195 \, L^{0.77} \, S^{-0.385} \qquad\qquad (5)$$

as in this empirical equation the time of concentration $t$ [min] is only related to the length of the main channel $L$ [m] and the slope of the longest hydraulic length $S$ [-]. Given the altitudes of Kharmong and Darband (upstream

Tarbela reservoir) gauging stations as 2,542 and 436 m a.s.l respectively and a channel length[1] of about 617 km, the approximated time-lag of 5000 min was finally rounded to 3 days.

## 2.4 Data sources

There are a number of high elevation climate stations in the Pakistani part of the Upper Indus Basin operated by WAPDA's[2] Glacier Monitoring and Research Centre (GMRC) and Pakistan Meteorological Department (PMD). However, they are concentrated on the western part of the UIB and data is not available online. In order to have most recent data for operational flow forecasting, the World Meteorological Organization (WMO) climate station at Srinagar airport located at an altitude of 1,587 m a.s.l. was chosen as temperature base station, which already had proven to give representative temperatures for that region in the SRM model of the Upper Jhelum catchment (Bogacki and Ismail, 2016) and a full set of climatic data can be obtained online from the GSOD[3] data-base with a time-lag of about 2 days only. Based on the daily air temperature data, degree-days in each elevation zone were calculated using a constant temperature lapse-rate of -6°C km$^{-1}$.

The MODIS/Terra Snow Cover Daily L3 Global 500 m Grid (MOD10A1) product[4] has been used to determine the daily snow covered area in the elevation zones. The compatibility of using MODIS data in conjunction with SRM in the Himalayas and its surroundings has already investigated by Immerzeel et al. (2009, 2010). As the MODIS sensor cannot detect snow below clouds, a cloud elimination algorithm is applied using temporal interpolation between two cloud-free days for each pixel. Afterwards the daily percentage of snow cover area in each elevation zone is calculated and smoothed by moving average.

At the beginning of the melting season, glaciers are usually completely covered by fresh snow. As the melting season progresses the snow cover will fade away and glacier exposed area will increase. The actual glacier extent was derived from two data sources. As a major source on global glacier distribution the Global Land Ice Measurements from Space (GLIMS) data archive was used (Raup et al., 2007). This data was complemented by interpretation of Landsat 8 scenes (30 m spatial resolution) from late summer to early fall 2013, in order to identify the maximum of the glacier exposed area. The merged data was mapped on the 500 m MODIS grid. On a daily basis, the glacier exposed area is determined by all pixels that are classified as glacier but not identified as snow by the MODIS sensor.

A spatial interpolation of in-situ (station) precipitation data in mountainous regions is particularly difficult and often biased towards lower values (Archer and Fowler, 2004) as the rain gauge network is usually sparse and mainly located at the valley floors while maximum precipitation occurs on mountains slopes and increases with altitude in general. A promising alternative to station data are gridded, remote sensing based precipitation products. However, regional and temporal pattern as well as multiannual means of these products differ significantly in the Himalayas (Palazzi et al., 2013). In particular, the widely used TRMM data-set is known to underestimate the precipitation in high altitudes as found in the UIB (Forsythe et al., 2011) or the Andes (Ward et al., 2011).

---

[1] digitised from Esri's World Imagery. Source: Esri, DigitalGlobe, GeoEye, i-cubed, USDA, USGS, AEX, Getmapping, Aerogrid, IGN, IGP, swisstopo, and the GIS User Community
[2] Pakistan Water and Power Development Authority
[3] Global Summary Of the Day. Download at: ftp://ftp.ncdc.noaa.gov/pub/data/gsod/
[4] Hall et al. (2006), updated daily. MODIS/Terra Snow Cover Daily L3 Global 500m Grid V005, [Feb. 2000 – Sep. 2016, tiles h23v05 & h24v05]. NSIDC Boulder, Colorado USA. Download at: ftp://n5eil01u.ecs.nsidc.org/SAN/MOST/MOD10A1.005

Based on own precipitation product comparisons for the Upper Chenab catchment, the gridded RFE 2.0 Central Asia[5] daily rainfall product (Xie et al., 2002) is used in the present model. According to SRM's elevation band approach, the gridded data, having a spatial resolution of 0.1° latitude/longitude, is mapped to the respective elevation zones. For the period 2003 – 2015 the product yields a mean annual precipitation of 854 and 482 mm/a

for the Lower and the Upper UIB respectively, that reflects the significantly lower annual precipitation on the Tibetan plateau compared to the western Himalayas (e.g. Bookhagen and Burbank, 2010; Ménégoz et al., 2013). The RFE basin-wide annual mean of 701 mm/a lies well in the range of 675 ±100 mm/a derived for the whole UIB by Reggiani and Rientjes (2015).

## 2.5 Model parameters

The most important parameter of a temperature-index model which is controlling daily snow and glacial melt is the degree-day factor [cm °$C^{-1}$ $d^{-1}$], which transforms the index variable degree-day [°C d] into actual melt [cm $d^{-1}$].
In case of glaciers a constant degree-day factor of 0.70 [cm °$C^{-1}$ $d^{-1}$] as proposed by Schaper et al. (2000) was chosen, which also corresponds to degree-day factors reported from glaciers in the Himalayas at a comparable latitude (Hock, 2003), the approach for degree-day factors for snow is more elaborated. In a first step, optimal

degree-day factors were obtained for each elevation zone and year by diagnostic calibration, i.e. by achieving the best possible fit between simulated and observed hydrographs for each year. From this calibration exercise it appears, that degree-day factors are increasing by the time after melting has started in a particular elevation zone (Figure 4 and 5). As a generalised rule is needed in the forecasting procedure, zone-wise degree-day factor functions as suggested by Ismail et al. (2015) were developed by linear regression between the calibrated degree-day factors

and time. The increase of the degree-day factors with the passage of time is because the snow absorbs energy due to its physical condition, in terms of increasing temperatures and solar radiations intensities. This process of energy storage plays a pivotal role in the ripening of the snowpack, which melts rapidly as the snow melting season progresses. The extent to which degree-day factors increase is related to the calibration procedure because it was observed during the model calibration that in a certain elevation zone when the degree-day factors attain the value

e.g (0.80 cm °$C^{-1}$ $d^{-1}$), the snow cover area in that very elevation zone has almost completely faded away so there is no need to further increase the values of degree-day factors. The limit to what extent the degree-day factors increase at a certain spatio-temporal region depends upon various physiographic and climatic parameters and a research is on-going for evaluating the trend of degree-day factors in response to the aforementioned parameters.
The start of snowmelt and correspondingly application of the developed degree-day factors generalised rule, is

correlated with a certain threshold temperature for each elevation zone (see Table 1 and 2). The other model parameters required by SRM like temperature lapse-rate, recession coefficient, runoff coefficient for snow, lag-time, etc., were applied basin-wide and kept constant for all years (see Table 3). The values of these parameters were determined according to the methods described by Martinec et al., (2011) and slightly adjusted to achieve a good fit over the whole calibration period. It has to be noted, that these parameter values will differ for other

catchments.

---

[5] RainFall Estimates version 2.0 created by the NOAA Climate Prediction Center's FEWS-NET group sponsored by USAID. Download at: ftp://ftp.cpc.ncep.noaa.gov/fews/afghan

## 2.6 Scenario approach for forecasting

In the forecasting period which starts from the 1$^{st}$ of April, the four model variables temperature, precipitation, snow covered area and glacier exposed area have to be predicted for the forthcoming 6 months of the Kharif cropping season (April – September). As the level of skill of seasonal climate forecasts for the Hindukush –
Karakoram – Western Himalaya region for such a lead time is still not sufficient, a scenario approach already successfully applied in the Upper Jhelum catchment (Bogacki and Ismail, 2016) is used.

This scenario approach has a lot in common with traditional Ensemble Streamflow Prediction ESP developed at the U.S. National Weather Service as a method for generating long-term probabilistic streamflow outlooks (Day, 1985). Based on the assumption that past meteorology is representative of possible future events, ESP uses
historical temperature and precipitation time series as forcings for the hydrological model to produce an ensemble of streamflow traces. A probabilistic forecast is created by statistical analysis of the multiple streamflow scenarios produced (Franz et al. 2008). Initial basin conditions are usually estimated by forcing the hydrological model with observed meteorology in a "warm-up" phase up to the time of forecast (Wood and Lettenmaier, 2008).

The seasonal scenario approach also uses historical temperature and precipitation as forcings for the SRM+G
model. In contrast to ESP however, this approach is, like the other operational forecast models for UIB, primarily focussed on a deterministic forecast of total Kharif flow volume. Besides the "most likely" (median) flow, SRM+G forecasts only give an indication of the bandwidth of expected flows by the dry (20%) and wet (80%) quantiles as limits of the "likely" range.

The other notable differences are the initial basin conditions. SRM and SRM+G do not use any initial conditions,
like soil moisture state of snow-water equivalent as used in other hydrological models. Instead however, the snow-cover area and the glacier exposed area are input variables to the model. For reasons of simplicity, the glacier exposed area is treated like the meteorological variables, i.e. the historical time-series are used. The depletion of the snow-covered area during the forecast period, which is the decisive factor for each forecast, is however predicted by so-called "modified depletion curves". These modified depletion curves are derived from the
conventional depletion curves of each elevation zone by replacing the time scale with the cumulative daily snow-melt depth (Martinec et al., 2011). The decline of the modified depletion curves depends on the initial accumulation of snow and represents the actual snow-water equivalent. When initial snow depth is low the modified depletion curve declines faster than in years when a lot of snow has accumulated. In the end of March, when the seasonal forecast is carried out, an elevation zone showing already some decline in snow covered area, and hence having
also some cumulated degree-days, is chosen as "key zone". Comparing the relation of decline in snow covered area versus cumulated degree-days with a statistical analysis of the modified depletion curves of previous years, the actual amount of snow is estimated and the future depletion anticipated accordingly, while assuming similar snow conditions for all elevation zones.

The major difference to other hydrological models as used in the ESP is the positive effect that usually the
uncertainty in the initial conditions is progressively superseded by the actual meteorological conditions. In SRM+G however, if an erroneous depletion estimate is in effect then it will persist during the whole forecast period. As all ensemble traces are based on the chosen depletion curves, the initial estimate is crucially influencing each trace of the ensemble in the same direction.

## 2.7 Verification methods

Model verification comprises the simulation model as well as the forecasting model. The accuracy of the simulation model was evaluated by the two standard criteria used in SRM (Martinec et al., 2011), namely the relative volume difference equation (6)

$$D_v = \frac{V - V^*}{V} \times 100 \ [\%] \tag{6}$$

and the coefficient of determination $R^2$ equation (7)

$$R^2 = 1 - \frac{\sum_{i=1}^{n}(Q_i - Q_i^*)^2}{\sum_{i=1}^{n}(Q_i - \bar{Q})^2} \tag{7}$$

where $V$ and $V^*$ are the observed and the simulated annual flow volumes, $Q_i$ and $Q_i^*$ are the observed and the simulated daily discharge values, and $\bar{Q}$ is the average observed daily discharge.

The skill of the forecasting model was assessed in comparison with IRSA's forecasts that are based on a statistical model and with forecasts from the UBC[6] watershed model (Quick and Pipes, 1977) that is used by WAPDA's Glacier Monitoring Research Centre. The set of verification metrics was chosen taking into account that the existing operational forecasts for Kharif flows are traditionally issued in form of deterministic forecasts, thus only the 'most likely' values forecasted by these models are available.

The accuracy of a forecast is a measure of the error between predicted and observed values. The root mean squared error RMSE equation (8), mean percentage error MPE equation (9), and mean absolute percentage error MAPE equation (10)

$$RMSE = \sqrt{\frac{1}{n}\sum_{i=1}^{n}(f_i - o_i)^2} \tag{8}$$

$$MPE = \frac{1}{n}\sum_{i=1}^{n}\frac{(f_i - o_i)}{o_i} \tag{9}$$

$$MAPE = \frac{1}{n}\sum_{i=1}^{n}\frac{|f_i - o_i|}{o_i} \tag{10}$$

were used as deterministic metrics to assess the accuracy of the predicted mean Kharif flow volumes. In the above equations, $f_i$ is the forecasted and $o_i$ the observed flow volume and $n$ the total number of considered forecasts. Both, RMSE and MAPE measure the average magnitude of the forecast errors, where RMSE penalises larger errors more than MAPE. The mean percentage error measures the deviation between average forecasted and average observed flows, i.e. a positive MPE indicates over-forecasting and a negative under-forecasting respectively.

As a commonly used deterministic measure of association, the correlation coefficient, equation (11)

---

[6] University of British Columbia Watershed Model

$$R = \frac{\sum_{i=1}^{n}(f_i - \bar{f})(o_i - \bar{o})}{\sqrt{\sum_{i=1}^{n}(f_i - \bar{f})^2}\sqrt{\sum_{i=1}^{n}(o_i - \bar{o})^2}} \tag{11}$$

was applied to assess the correspondence between forecasted and observed values. In addition, the uncentered anomaly correlation $AC_u$ (Wilks, 2006) equation (12)

$$AC_u = \frac{\sum_{i=1}^{n}(f_i - \bar{c})(o_i - \bar{c})}{\sqrt{\sum_{i=1}^{n}(f_i - \bar{c})^2}\sqrt{\sum_{i=1}^{n}(o_i - \bar{c})^2}} \tag{12}$$

where $\bar{c}$ is the climatological average value, was used as another measure of association. The anomaly correlation is designed to measure similarities in the patterns of anomalies from the climatological average between forecasted and observed values. An AC $\geq 0.6$ is usually regarded as an indication of some forecasting skill (Wilks, 2006). In the present context, the climatology average $\bar{c}$ is equivalent to the average observed flows $\bar{o}$.

The ability of a non-probabilistic forecast to predict extreme conditions is usually assessed by defining discrete
categories like below normal, normal, and above normal. The Heidke and Peirce skill scores for multi-categorical forecasts measure the fraction of correct forecasts in each category in relation to those forecasts which would be correct due purely to random chance. The Peirce skill score, equation (13)

$$PSS = \frac{\sum_{j=1}^{m} p(f_j, o_j) - \sum_{j=1}^{m} p(f_j)\, p(o_j)}{1 - \sum_{i=j}^{m} p(o_j)^2} \tag{13}$$

is unbiased in the sense that it assigns a marginal distribution to the reference random forecast which is equal to the (sample) climatology (Wilks, 2006). In the above equation, $m$ is the number of categories, $p(f_j, o_j)$ the joint distribution of forecasts and observations, where $p(f_j)$ and $p(o_j)$ are the respective marginal distributions.

As the existing operational forecasts are primarily designed as point estimates of the mean flow volume and hence
a comparison of probabilistic metrics between these models is not possible, only a basic probabilistic evaluation of the SRM+G scenario ensembles was carried out. The ranked probability score RPS, which is essentially an extension of the Brier score to the many-event situation (Wilks, 2006), was used. As it reflects the overall performance of a multi-category probabilistic forecast (Franz et al., 2003). In order to calculate the RPS, first the quantiles of $m$ categories have to be determined based on given non-exceedance probabilities of the observed
values. Then, for each forecast the ensemble members as well as the observed flow are assigned to these categories and the respective cumulative distributions, equation (14)

$$F_i = \sum_{j=1}^{m} p_j(f_i) \qquad O_i = \sum_{j=1}^{m} p_j(o_i) \tag{14}$$

are calculated, where $F_i$ is the cumulative ensemble distribution of forecast $i$ and $p_j(f_i)$ the relative frequency of an ensemble member falling into category $j$. For each forecast $i$ there is only one observation $o_i$, hence the category
$j$ the observation falling in is given a relative frequency of $p_j(o_i) = 1$ while all others are set to 0. Finally, the RPS, equation (15) for $n$ forecasts is the average of the sum of the squared differences of the cumulative distributions.

$$RPS = \sum_{i=1}^{n} \left\{ \sum_{k=1}^{m} \left[ \sum_{j=1}^{k} p_j(f_i) - \sum_{j=1}^{k} p_j(o_i) \right]^2 \right\} \qquad (15)$$

The RPS penalises forecasts more severely when their probabilities are further from the actual observations. The relative improvement or skill of a probability forecast over climatology as a reference forecast is assessed by the ranked probability skill score RPSS, equation (16)

$$RPSS = 1 - \frac{RPS}{RPS_{ref}} \qquad (16)$$

where $RPS_{ref}$ is the RPS calculated with a constant forecast, e.g. the average of the observed series.

Besides the RPSS as a single-number score for the forecast performance, the reliability diagram (Wilks 2006) is used to show the full joint distribution $p(f_i, o_j)$ of forecasts and observations of a binary predictand in terms of its calibration – refinement factorisation (Murphy and Winkler 1987)

$$p(f_i, o_j) = p(o_j|f_i) \cdot p(f_i) \qquad i = 1, \dots, m \;\; j = 1, \dots, n \qquad (17)$$

where the $m$ conditional distributions $p(o_j|f_i)$ specify how often each possible observation $o_j$ occurred when the particular forecast $f_i$ was issued, or in other words how well each forecast $f_i$ is calibrated. Forecasts $f_i$ that fall near to the 1:1 line in the reliability diagram result in a small (good) reliability term of the algebraic decomposition of the Brier score (Murphy 1973), which is the weighted average of the squared vertical distances.

15  The other part of the above factorization is the unconditional (marginal) distribution $p(f_i)$ that specifies how often each of the $m$ possible forecast values occurred. This so-called refinement distribution is visualised by a probability histogram which also is referred to as sharpness diagram. A distribution with a large spread indicates that different forecasts are issued relatively frequently, and so have the potential to discern a broad range of conditions. Conversely, a narrow distribution, i.e. if most of the forecasts $f_i$ are the same or in a similar range, indicates a lack

20  of sharpness (Wilks 2006).

While the calibration – refinement factorisation relates to a binary predictand, the SRM+G scenario forecasts are grouped in three categories, i.e. less than normal, near normal (most likely), and higher than normal. According to Murphy (1972) this $N$-state situation is handled as a collection of $N * m$ *scalar* forecasts, thus treating each category as a separate binary forecast $f_i$ that either meets or not meets the observation $o_j$.

## 3 Results and discussion

The development of the SRM+G forecasting model for the UIB has been an iterative process with the focus on creating an operational forecasting tool for Kharif flow volumes to the Tarbela reservoir. Thus not all improvements have been tested individually while not changing the other components which would allow an independent

30  assessment of the individual effects. Nevertheless, the results are discussed below separately for each component.

## 3.1 Splitting of the UIB catchment

While the simulation results using a sole model for the whole UIB showed an acceptable agreement between simulated and observed flows in terms of $R^2$ and $D_v$, initial hindcast results proved to be not satisfactory, especially

for the Early Kharif (1$^{st}$ April – 10$^{th}$ June) season, which is the major snowmelt contribution period. The mean percentage error MPE between hindcasts and observations of -21.0% for the years 2003 – 2014 indicated a severe bias towards under-estimating the actual flows and the respective mean absolute percentage error MAPE of 25.9% was also unexpectedly large.

An analysis of MODIS snow cover data indicates that in the south-eastern part of the UIB namely the Tibetan plateau already in March the snow cover is fading away rapidly while on the other hand, in the north-western part of the catchment the same elevation zone is still widely covered with snow (Figure 6). In Table 4 the snow cover area of the relevant elevation zones for the south-eastern (Upper) and north-western (Lower) part of the UIB is given on 1$^{st}$ March and 1$^{st}$ April as an example for the year 2003. While at an elevation of 4,000 m a.s.l. the snow
cover area reduces from 82% to 71% in the Lower UIB, in the Upper UIB the snow cover area shrinks sharply from 79% to 50%. A similar behavior can be observed for most of the other years as well.

As in forecasting mode the depletion of the snow cover area during the whole forecasting period is predicted depending on the reduction in the "key zone" in March (see Chap. 2.6), the relatively larger depletion in the Upper UIB leads to an under-estimation of the available snow-water equivalent for the whole catchment, which explains
the subsequent under-estimation of Early Kharif flows by the initial hindcasts and in turn motivates the splitting of the UIB into two sub-catchments and separate models respectively (see Chap. 2.3). As a result of this splitting, the MPE of the hindcasts for Early Kharif changed to a modest over-estimation of 4.2% while the MAPE could be reduced to 15.8%.

## 3.2 Glacier melt component

During a first diagnostic calibration of the degree-day factors it became obvious, that in late summer even with extreme high degree-day factors it was usually not possible to reproduce the observed hydrograph. The analysis of the snow cover depletion showed, that in most of the years snow has vanished from areas below 4,000 m a.s.l. already in June and elevation zones below 5,000 m a.s.l. usually become snow-free in July. Thus, the snow-melt contribution to the flow is rapidly diminishing in August and September (see Figure 7).

Although in July usually the monsoon season starts bringing the highest monthly precipitation depth to the UIB in July and August, the resulting flow component from rain is not sufficient to create the necessary discharge. Therefore, many studies postulate a substantial contribution from glacier melt to the annual flow in the UIB, e.g. Immerzeel et al. (2010) estimate a contribution of 40% from snow and 32% from glacier melt with the remaining 28% from rain, Charles (2016) mentioned that the contribution is 50% from snow- and 20% from glacial melt.

Figure 7 shows the monthly distribution of the three flow components as calculated by SRM+G before subjecting to the recession flow calculation according to eq. (1). SRM's simple recession flow approach is not mass conservative and does also not allow a direct attribution, at what day these flow components actually occur in the daily discharge $Q_{n+1}$. However, an overall water balance shows, that the difference between water going into the (virtual) storage and water taken out by the recession flow term $Q_n$ is about 7% which seems acceptable in relation
to the uncertainty associated with the input data. Having in mind the above limitation, the average (2003 – 2014) flow component distribution as simulated by SRM+G is 53%, 21%, and 26% for snow, glacier and rain respectively, which is well in the magnitude of the values found in other studies.

Figure 8 and 9 compare the hydrographs of simulation runs with and without the glacier component for Upper and Lower UIB. The effect is more visible in the Lower UIB as about 10.5% of the catchment is glaciated, while for the Upper UIB the glaciated area is merely 1.7%.

### 3.3 Simulation model verification

The simulation model was verified by comparing full year (1st January – 31th December) simulation runs using the actual temperature, precipitation and snow cover data versus observed daily flows. Examples of respective hydrographs for Upper and Lower UIB are given in Figure 10 and 11. The resulting coefficients of determination $R^2$ and relative volume differences $D_v$ for each year and both, the Upper and Lower UIB model, are given in Table 5. Although the years 2003 – 2012 were used to calibrate certain model parameters (see Chap 2.5), they are regarded

to validate the model in "forecasting mode", as in particular the degree-day factor functions (Table 1 and 2) were applied as during a real forecasting procedure, i.e. the starting point was selected according to Tables 1 and 2 depending on the historic 10-days temperatures for each year. The years 2013 and 2014 on the other hand were not used at the time of model calibration; thus they represent a fully independent verification of the simulation model. The average $R^2$ of 0.86 and 0.88 for the Upper and Lower UIB respectively indicate that in general the two models

simulate the variations of the observed hydrographs quite acceptable. Values for the years 2013 and 2014 are even above the average for Lower UIB, which is finally essential for Tarbela inflows, but extreme floods, like 2010 during monsoon season, are reproduced not that well. The average relative volume differences $D_v$ of -1.89% and 0.03% for Upper and Lower UIB respectively show, that the simulation models, although not mass-conservative due to the recession approach (Chap. 2.2) and volume differences vary from year to year, are in terms of total flow

volume not biased over the long run.

### 3.4 Evaluation of forecasting skills

In order to evaluate the skills of the forecasting model, hindcasts were carried out for the years 2003 – 2014 using always all years, i.e. also the hindcasted year, as scenario members. For the years 2015 and 2016 real forecasts have been determined before 1st April of these years, thus without using the particular year as a scenario. In all

cases, the expected depletion of snow cover area was predicted for each scenario member based on the respective situation in March of the specific year.

In Table 6 the ensemble medians of hind- / forecasts by SRM+G are compared with observed flows as well as with IRSA's forecasts that are based on a statistical model and with forecasts from the UBC watershed model (Quick and Pipes, 1977) that is used by WAPDA's Glacier Monitoring Research Centre. All values are total Kharif (1st

April – 30th September) flow volumes in $10^9$ m$^3$. Figure 12 presents all model results and observed flows and shows also the 20% and 80% quantiles of the SRM+G scenario ensemble. Table 7 summarises the metrics that are used to compare the forecast skills of the three models. The RMSE, MAE and MAPE show an improvement in accuracy by SRM+G and the MPE a reduction of bias where SRM+G tends to slightly under-estimate, while the two other forecasts moderately over-estimate the total Kharif flows. Both, the correlation coefficients $R$ and the anomaly

coefficients $AC_u$ indicate however, that the association between forecasts and observations is weak for all three models. Here, the UBC forecasts show the best correlation followed by SRM+G and IRSA. The above aspects of model performance are synoptically visualised in the Taylor diagram (Taylor, 2001) in Figure 13 that was plotted

using the R-package Plotrix (Lemon, 2006). All models are comparable far away from the point of observations given on the x-axis, with SRM+G having the smallest centered root mean square difference and UBC the best correlation coefficient.

In order to evaluate the model skills in forecasting extreme, i.e. dry or wet, conditions, the Peirce skill score was applied. The limits between the categories dry, normal, and wet conditions were defined as 20% and 80% non-exceedance of the observed historic Kharif flow series 2003 – 2016, which corresponds to quantiles of 56.8 $km^3$ and 67.9 $km^3$ respectively. Obviously IRSA and SRM+G forecasts have no skill in this respect, while UBC shows some, however limited skill compared to purely random chance.

As only point estimates of IRSA and UBC forecasts were available, the assessment of probabilistic skill only applies to the SRM+G scenario ensembles. Figures 14 and 15 show typical traces of ensemble members as well as the observed, mean, and historic trace. In most years the ensemble mean is closer to the observed value than the historic trace.

The ranked probability skill score RPSS was used to assess the overall performance of the probabilistic forecast. Same category limits as for the Peirce skill score were chosen, i.e. the 20% and 80% percentiles of the observed flow series. As no other probabilistic forecasts were available as reference forecast, the ranked probability score RPS of the scenario forecast ensembles was compared to the climatology, i.e. the average Kharif flow of the observed series of 62.7 $km^3$. The RPS of scenario forecast ensembles and climatology were 0.370 and 0.462 respectively. The resulting RPSS of 0.20 indicates that the scenario ensemble shows some, however limited skill and improvement over a constant forecast of the historic average.

The reliability diagram (Figure 16) was constructed with the aforementioned three forecast categories dry ($\leq 56.8$ $km^3$), near normal, and wet ($> 67.9$ $km^3$) conditions based on the historic observations and using for each forecast the frequency of ensemble members falling into the respective category. The resulting reliability diagram is relatively rugged and has gaps in several classes, although each forecast category was treated as individual forecast resulting in 42 scalar forecasts. However, in total there are only 14 observations which e.g. causes the outlier at the forecast probability class 0.6. There were 3 forecasts fall into this very class but none out of the 14 observations. As it can be seen from the sharpness diagram, there are also empty forecast classes 0.4 and 0.5 meaning these probabilities have never occurred in any forecast category. Besides the shortcoming of the limited sample size and in particular the outlier, most points are located around the 1:1 line, indicating there is no dry or wet bias, and they fall within an acceptable distance. Resolution, although difficult to assess taking the limited number of points, seems to be a bit weak, i.e. with a tendency to over-confidence, which may be caused by the fact that the available scenario years comprise mostly near normal conditions.

## 4 Conclusions

The snowmelt runoff model, which in combination with a scenario approach is successfully applied to predict seasonal flows e.g. in the Upper Jhelum catchment, was applied to the Upper Indus Basin in order to forecast the total Kharif inflow to the Tarbela reservoir. Several improvements had to be introduced to SRM in order to meet the specific requirements of the UIB.

Not surprisingly, a separate component had to be added to SRM in order to consider the flow component originating from glacier melt. Without this component, especially in the late summer months there is a lack of water to meet the observed hydrograph. All year simulation runs with SRM+G for the period 2003 – 2014 result in an average flow component distribution of 53%, 21%, and 26% for snow, glacier and rain respectively, which fits well to the

values found in a number of other studies.

It is well known, that the Tibetan plateau receives significantly lesser precipitation than the western parts of the UIB. In addition, MODIS data shows that the snow cover is fading away in early spring much faster than in the other parts. In the present study, SRM's modified depletion curve approach for predicting the snow cover depletion during the forecast period has proven to be very sensitive to errors in the estimation of the actual snow-water

equivalent. In such cases it is inevitable to split the catchment into more homogeneous units. Therefore, the superposition of flows from sub-catchments by using a time-lag was introduced to SRM+G, which leads to a significant improvement in the forecasts of snow-melt dominated Early Kharif flows in particular.

The scenario approach is a step towards probabilistic forecasting of seasonal flows in the UIB. As the accuracy of existing forecasts with a mean volume error of 10.9% and 11.4% is already quite high, the improvement by SRM+G

having a MAPE of 9.5% is only limited. The bias however could be reduced to -2.0%. Association between forecasts and observations is rather weak for all three models, just as none of the models has significant skill in predicting extreme dry or wet conditions.

Regarding the scenario approach it is obvious, that as far as the variables precipitation and temperature are concerned, these tend toward the climatology, i.e. the long term averages. A variance in the forecasts is only

introduced by the different estimates of the snow-cover depletion curves for each forecast. Thus a promising way to improve the association and sharpness of the scenario approach would be a selection of a subset of ensemble members according to forecasted seasonal anomalies in temperature and precipitation. A quick test using only the five lowest, middle, or highest ensemble members selected according to the (known) relative flow frequency of the forecasted year gives promising results, e.g. not only a MAPE of 4.9%, but also an $AC_u$ of 0.78 and a PSS of 0.41.

The challenge of course is to forecast the seasonal anomaly in temperature and precipitation. In this respect further research is needed on how today's global forecast systems may allow a more specific selection of ensemble members particularly in the UIB, where the correlation to common teleconnections like the ENSO[7] status is known to be weak.

**Acknowledgements**

The authors thank National Engineering Services Pakistan (Pvt.) Ltd. (NESPAK), Lahore and AHT Group AG, Essen, Germany that they could be part of the project team. They are highly grateful to Indus River System Authority (IRSA) and WAPDA's Glacier Monitoring Research Centre (GMRC) and Surface Water Hydrology Project (SWHP) of for sharing their forecast results as well as the daily discharge data. The authors are also thankful

to the two anonymous reviewers for their valuable and constructive comments that substantially helped to improve the quality of the manuscript.

---

[7] El Niño Southern Oscillation

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

**Table 1: Zone-wise degree-day factors and 10-daily temperature threshold depending on 10-daily periods after melting start for Lower UIB**

| | Elevation Zone (m a.s.l) | | | | | | | |
|---|---|---|---|---|---|---|---|---|
| | 0-2500 | 3000 | 3500 | 4000 | 4500 | 5000 | 5500 | >5500 |
| 10-daily Period | 1-4 | 5 | 6 | 7 | 8 | 9 | 10 | 11 |
| $T_{10d}$[8] | 9.0 | 7.0 | 5.0 | 4.0 | 2.0 | 1.0 | 1.0 | 1.0 |
| 1 | 0.20 | 0.21 | 0.22 | 0.22 | 0.19 | 0.18 | 0.18 | 0.20 |
| 2 | 0.30 | 0.32 | 0.32 | 0.32 | 0.30 | 0.31 | 0.31 | 0.33 |
| 3 | 0.39 | 0.43 | 0.41 | 0.43 | 0.41 | 0.43 | 0.44 | 0.46 |
| 4 | 0.48 | 0.53 | 0.51 | 0.54 | 0.52 | 0.56 | 0.57 | 0.59 |
| 5 | 0.57 | 0.64 | 0.61 | 0.65 | 0.63 | 0.68 | 0.70 | 0.72 |
| 6 | 0.67 | 0.75 | 0.70 | 0.80 | 0.74 | 0.80 | 0.80 | 0.80 |
| 7 | 0.80 | 0.80 | 0.80 | 0.80 | 0.80 | | | |

**Table 2: Zone-wise degree-day factors and 10-daily temperature threshold depending on 10-daily periods after melting start for Upper UIB**

| | Elevation Zone (m a.s.l) | | | | | | |
|---|---|---|---|---|---|---|---|
| | 3000 | 3500 | 4000 | 4500 | 5000 | 5500 | >5500 |
| 10-daily Period | 1 | 2 | 3 | 4 | 5 | 6 | 7 |
| $T_{10d}$ | 2.0 | 2.0 | 2.0 | 2.0 | 0.5 | 0.5 | 0.5 |
| 1 | 0.37 | 0.35 | 0.35 | 0.52 | 0.56 | 0.48 | 0.60 |
| 2 | 0.43 | 0.40 | 0.40 | 0.59 | 0.64 | 0.54 | 0.70 |
| 3 | 0.49 | 0.45 | 0.46 | 0.66 | 0.73 | 0.80 | 0.80 |
| 4 | 0.54 | 0.51 | 0.51 | 0.73 | 0.80 | | |
| 5 | 0.60 | 0.56 | 0.56 | 0.80 | | | |
| 6 | 0.66 | 0.61 | 0.62 | | | | |
| 7 | 0.71 | 0.66 | 0.67 | | | | |

---

[8] 10-Daily average temperature in °C in each elevation zone.

**Table 3: SRM+G Model Parameters for both Upper and Lower UIB**

| Parameters | Symbol | Value | Units | Remarks |
|---|---|---|---|---|
| Temperature Lapse-Rate | $\gamma$ | 6.0 | °C km$^{-1}$ | |
| Recession Coefficient | $k_x$ | 1.193 | | October-February |
| | | 1.060 | | March – September |
| | $k_y$ | 0.029 | – | October-February |
| | | 0.020 | | March – September |
| Critical Precipitation | $P_{crit}$ | 1 | cm | constant |
| Lag Time | $L$ | 54 | h | 2.5 days delay between melt and runoff at Tarbela |
| Critical Temperature | $T_{crit}$ | 0.5 – 3.0 | °C | variable |
| Rainfall Contributing Area | $RCA$ | 0 | – | November – March |
| | | 1 | | April – October |
| Runoff Coefficient - Snow | $c_S$ | 0.80 | – | constant |
| Runoff Coefficient - Glacier | $c_G$ | 0.70 | – | constant |
| Runoff Coefficient - Rain | $c_R$ | 0.25-0.75 | – | |
| Degree-Day Factor - Snow | $\alpha$ | 0.15-0.80 | cm °C$^{-1}$d$^{-1}$ | |
| Degree-Day Factor - Glacier | $a_G$ | 0.70 | cm °C$^{-1}$d$^{-1}$ | constant |

**Table 4: Depletion of snow cover area for Upper and Lower UIB during March 2003**

| Elevation (m asl) | 3500 | 4000 | 4500 | 5000 | 5500 | >5500 |
|---|---|---|---|---|---|---|
| | | | 1$^{st}$ March | | | |
| Lower UIB | 66% | 82% | 88% | 87% | 83% | 94% |
| Upper UIB | 58% | 79% | 58% | 51% | 58% | 71% |
| | | | 1$^{st}$ April | | | |
| Lower UIB | 42% | 71% | 84% | 84% | 78% | 92% |
| Upper UIB | 24% | 50% | 48% | 43% | 51% | 73% |

**Table 5: Coefficient of determination R² and relative volume difference D_v for Upper and Lower UIB**

| Year | Upper UIB $R^2$ | Upper UIB $D_v$ [%] | Lower UIB $R^2$ | Lower UIB $D_v$ [%] |
|---|---|---|---|---|
| 2003 | 0.86 | -17.3 | 0.92 | 4.6 |
| 2004 | 0.84 | 2.2 | 0.90 | 0.1 |
| 2005 | 0.89 | 15.3 | 0.83 | -17.4 |
| 2006 | 0.85 | 8.4 | 0.91 | -3.5 |
| 2007 | 0.80 | 4.0 | 0.88 | -4.1 |
| 2008 | 0.94 | -6.7 | 0.92 | -1.4 |
| 2009 | 0.79 | 14.2 | 0.86 | 16.4 |
| 2010 | 0.90 | -1.9 | 0.77 | -16.3 |
| 2011 | 0.88 | -9.1 | 0.88 | 4.5 |
| 2012 | 0.87 | -16.0 | 0.89 | 11.9 |
| 2013 | 0.77 | -9.3 | 0.93 | -2.5 |
| 2014 | 0.88 | -6.5 | 0.92 | 8.0 |
| **Average** | **0.86** | **-1.89** | **0.88** | **0.03** |

**Table 6: Comparison of Kharif flow volumes [km³] 2003 - 2016**

| Year | Observed | IRSA | UBC | SRM+G |
|---|---|---|---|---|
| 2003 | 67.8 | 64.0 | 63.5 | 63.1 |
| 2004 | 51.8 | 60.5 | 63.6 | 60.8 |
| 2005 | 68.9 | 69.0 | 73.3 | 60.9 |
| 2006 | 67.8 | 68.4 | 73.3 | 61.6 |
| 2007 | 60.5 | 74.9 | 70.1 | 61.0 |
| 2008 | 57.7 | 68.5 | 59.2 | 53.9 |
| 2009 | 57.6 | 63.7 | 67.2 | 62.4 |
| 2010 | 76.6 | 63.3 | 68.4 | 61.4 |
| 2011 | 60.0 | 67.2 | 70.8 | 59.9 |
| 2012 | 55.4 | 61.3 | 61.7 | 60.4 |
| 2013 | 65.6 | 64.9 | 58.8 | 59.8 |
| 2014 | 52.9 | 64.6 | 64.2 | 61.4 |
| 2015 | 67.2 | 63.3 | 61.3 | 58.9 |
| 2016 | 66.4 | 62.4 | 66.5 | 63.1 |

**Table 7: Comparison of forecast skills between IRSA, UBC, and SRM+G**

| Model | MAE km³ | RMSE km³ | MPE % | MAPE % | R - | $AC_u$ - | PSS - |
|---|---|---|---|---|---|---|---|
| **IRSA** | 6.5 | 8.0 | 5.8 | 10.9 | 0.107 | 0.085 | -0.070 |
| **UBC** | 6.9 | 7.7 | 6.3 | 11.4 | 0.318 | 0.260 | 0.096 |
| **SRM+G** | 6.0 | 7.0 | -2.0 | 9.5 | 0.223 | 0.168 | -0.079 |

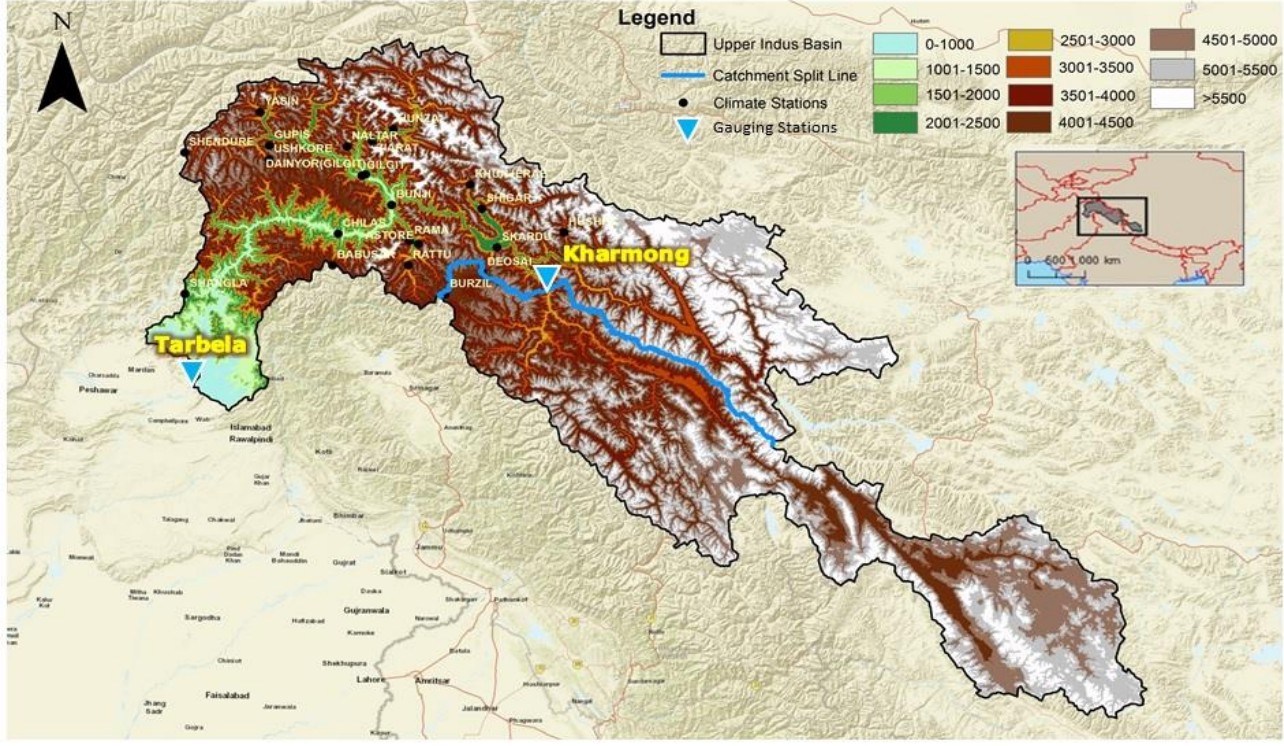

**Figure 1: Map of the Upper Indus Basin showing different elevations and splitting of UIB at the Kharmong gauging station into Upper and Lower UIB**

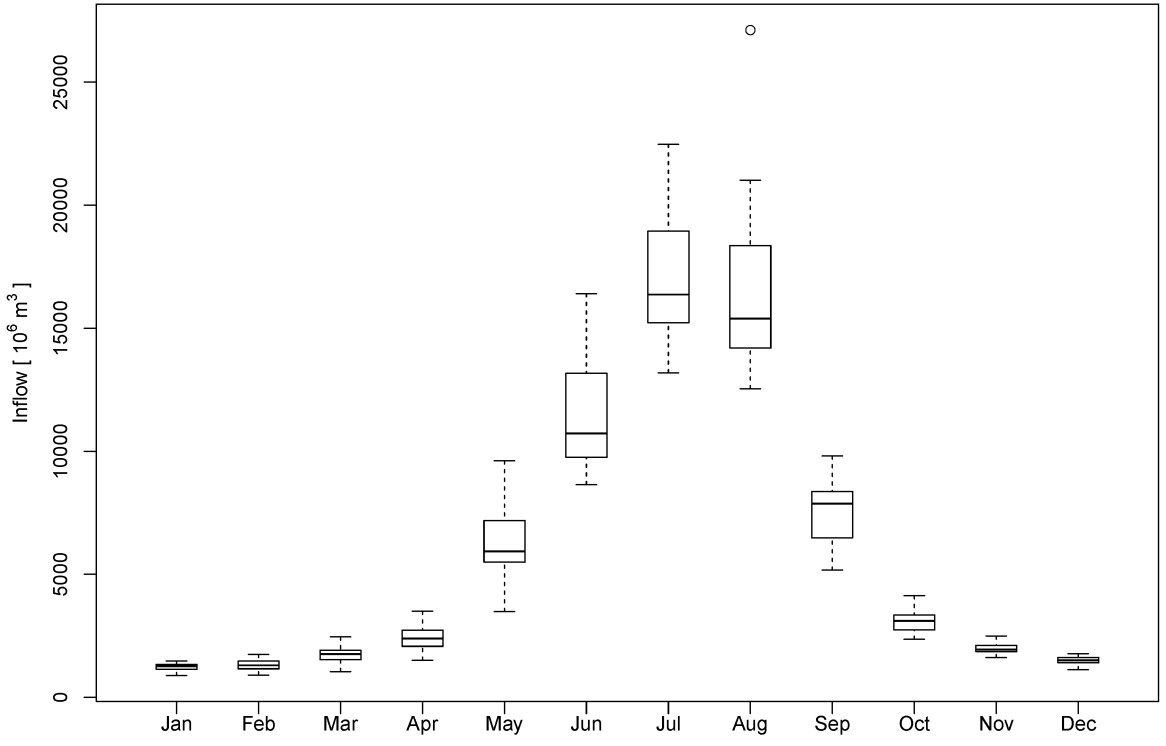

**Figure 2: Monthly distribution of inflows to the Tarbela Reservoir from 2000 - 2015**

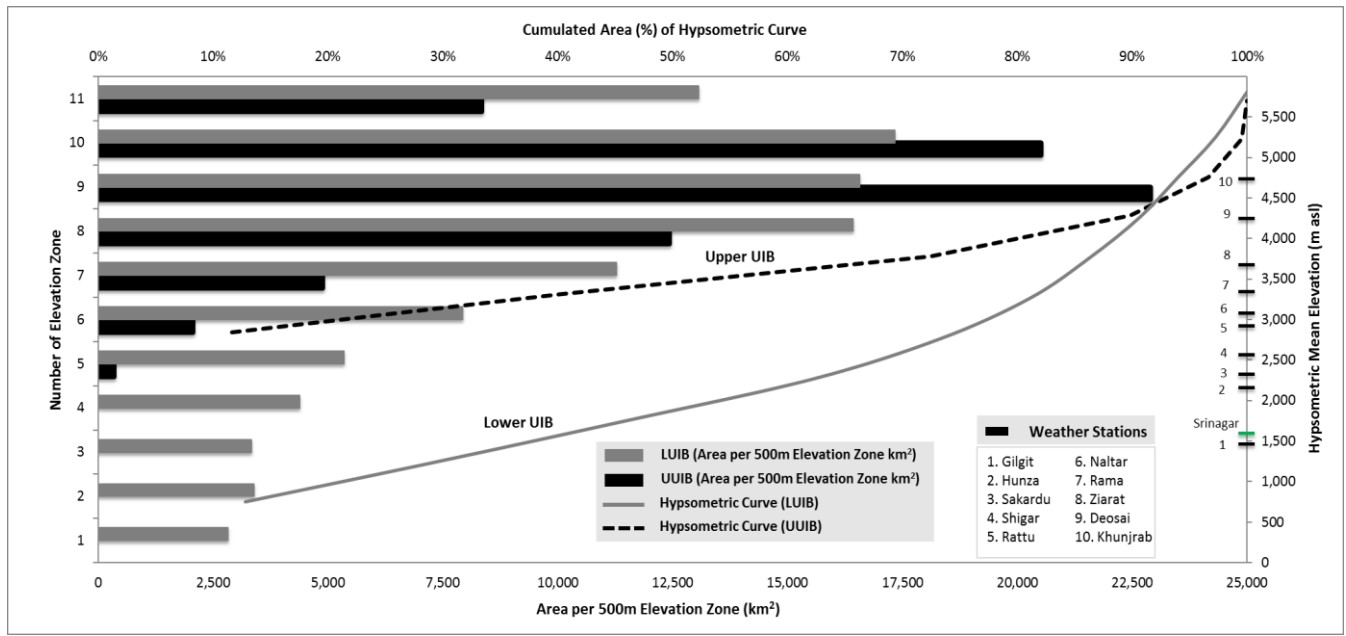

**Figure 3: Hypsometric curves and the distribution of area under 500-m elevation bands for the Upper and Lower UIB. Eleven as well as seven elevation zones were made for Upper and Lower UIB and the elevation of the weather stations in western portion of the UIB are presented on the right hand side y-axis.**

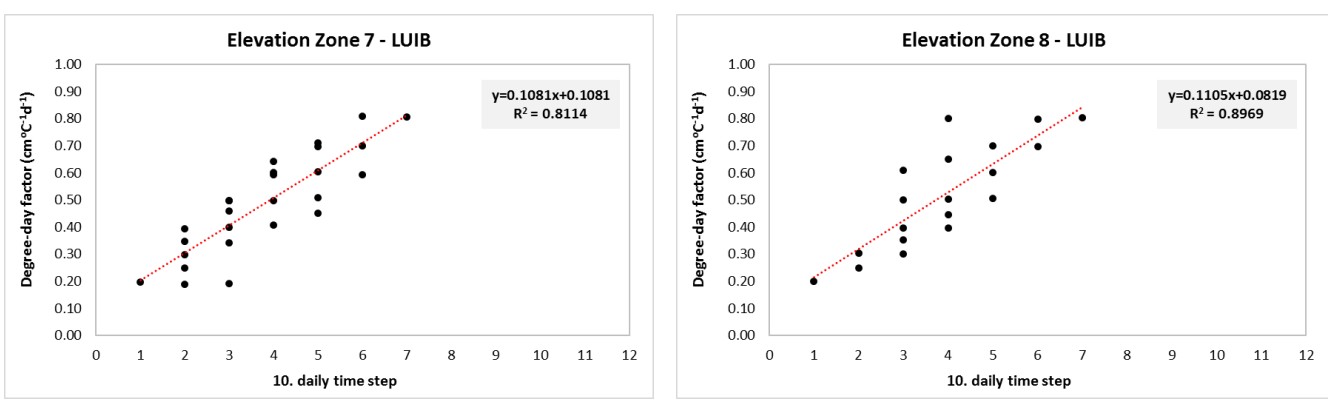

**Figure 4: Increase of degree-day factors with time (10-days periods) after melting start for elevation zones 7 and 8 for Lower UIB. Degree-day factors are obtained by diagnostic calibration.**

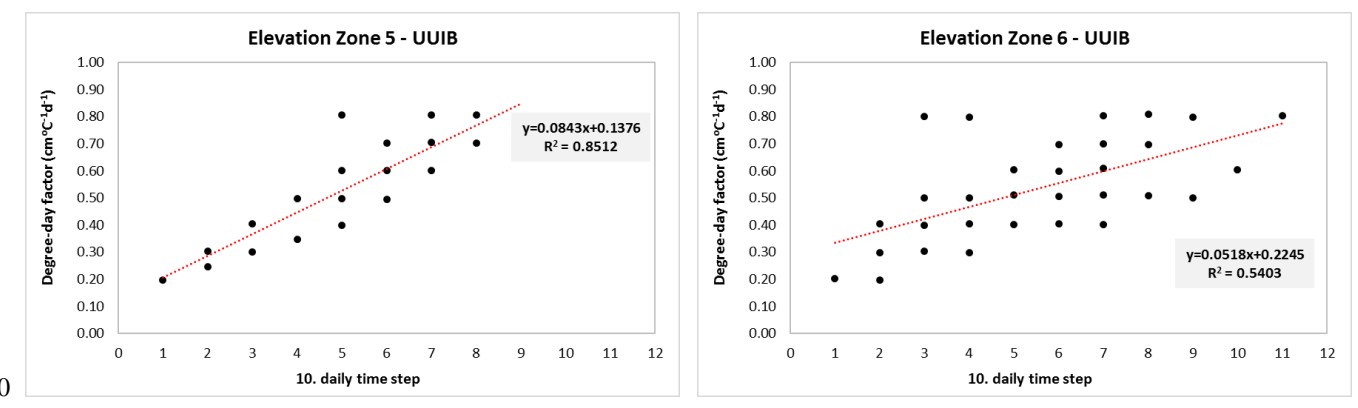

**Figure 5: Increase of degree-day factors with time (10-days periods) after melting start for elevation zones 5 and 6 for Upper UIB. Degree-day factors are obtained by diagnostic calibration.**

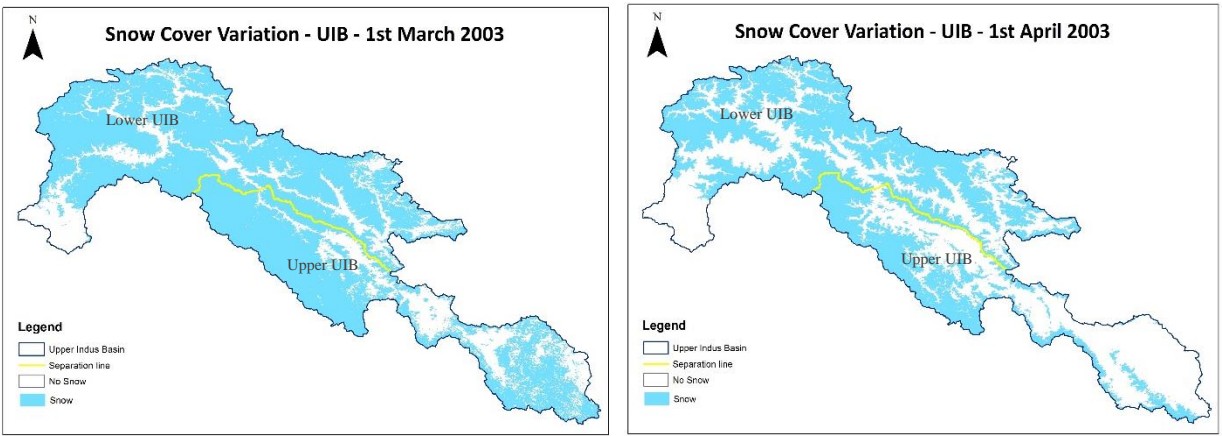

**Figure 6: Snow cover variation in the month of March and April 2003 in UIB**

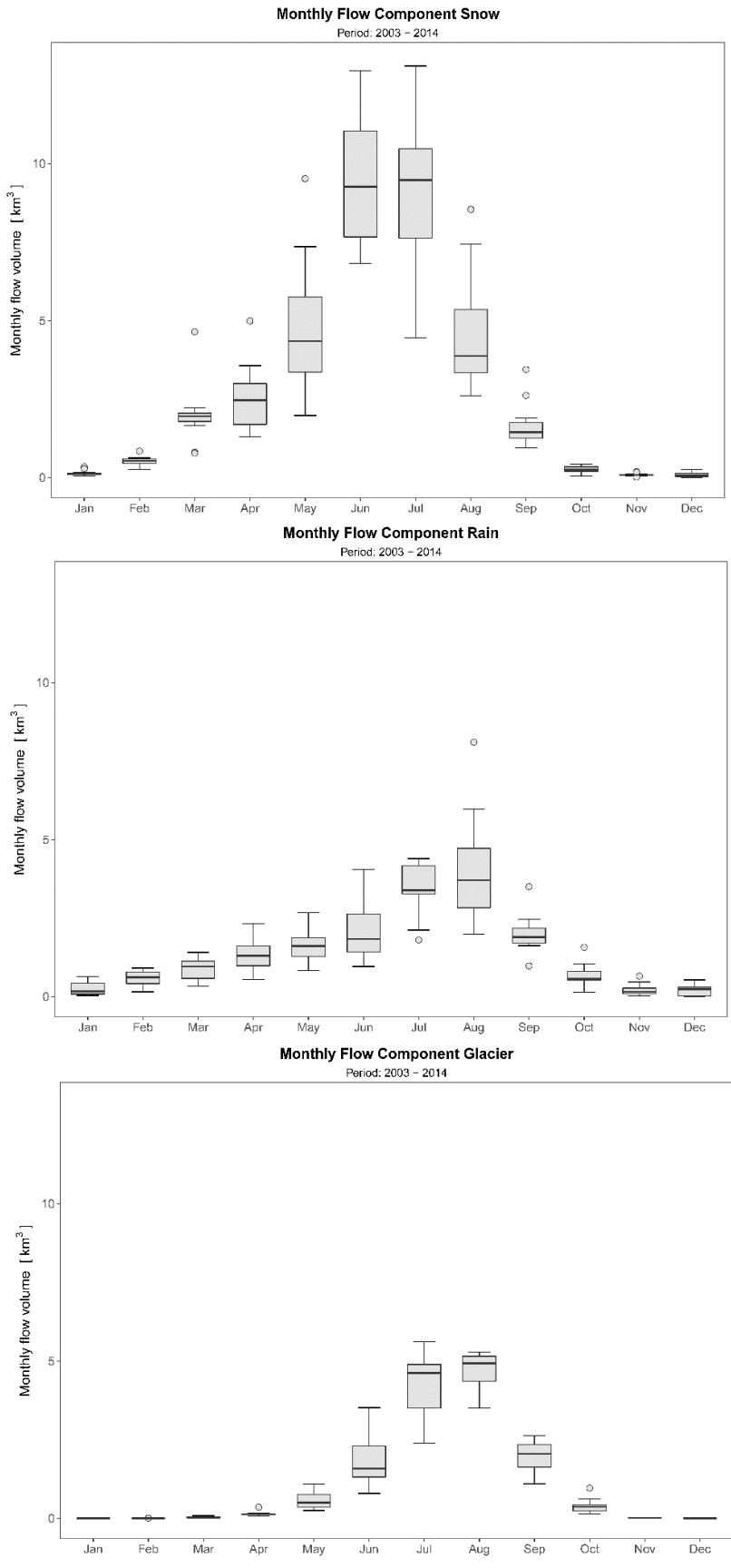

**Figure 7: Monthly distribution of flow components in UIB**

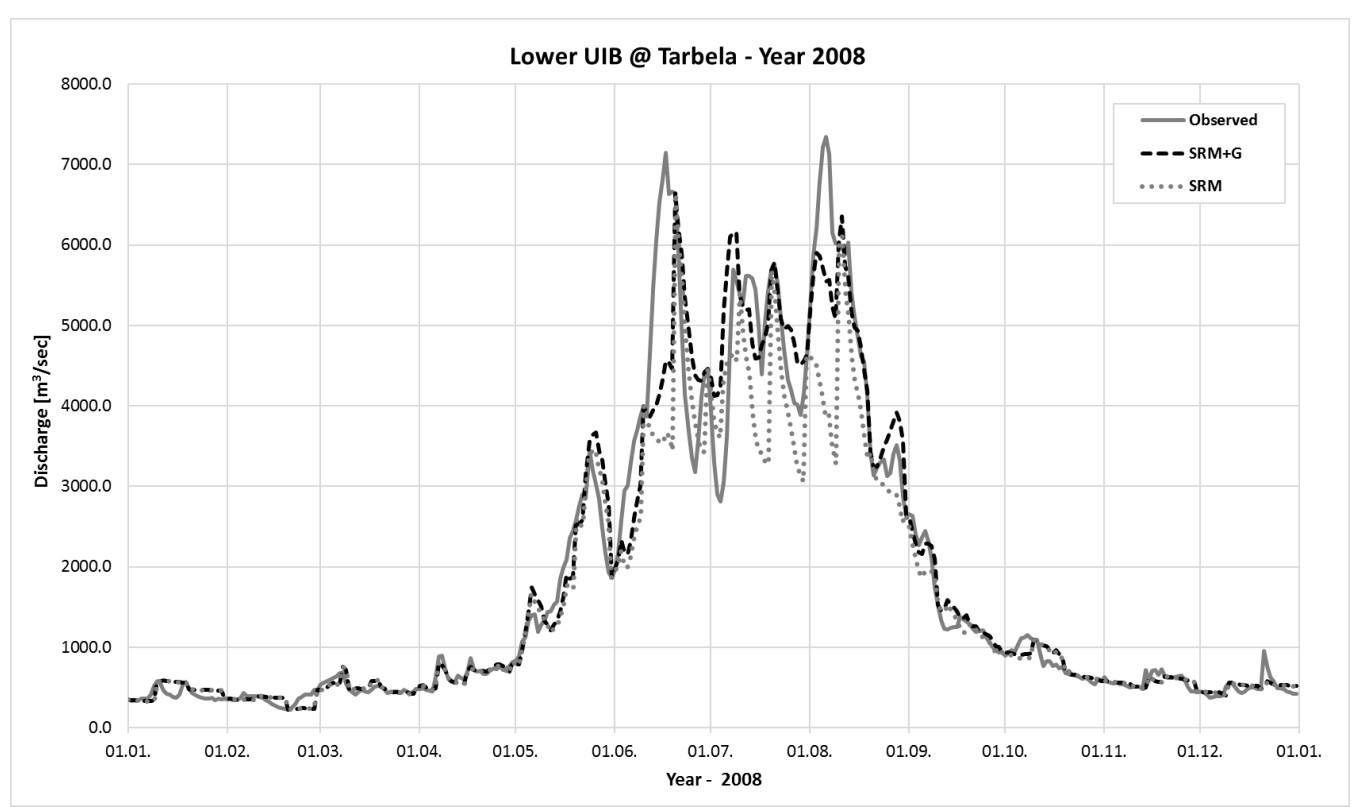

**Figure 8: Comparison of SRM+G (with glaciers) and SRM (without glaciers) for Lower UIB – 2008**

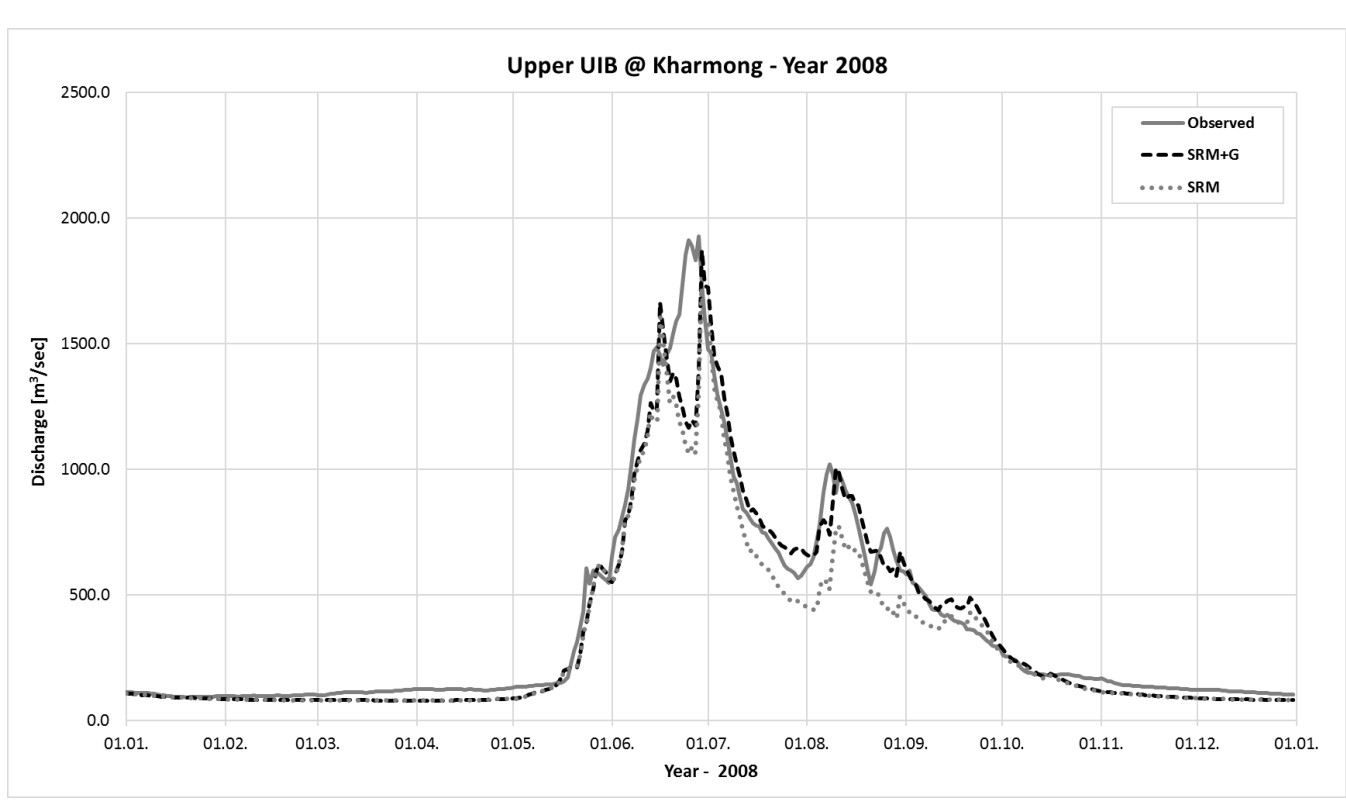

**Figure 9: Comparison of SRM+G (with glaciers) and SRM (without glaciers) for Upper UIB - 2008**

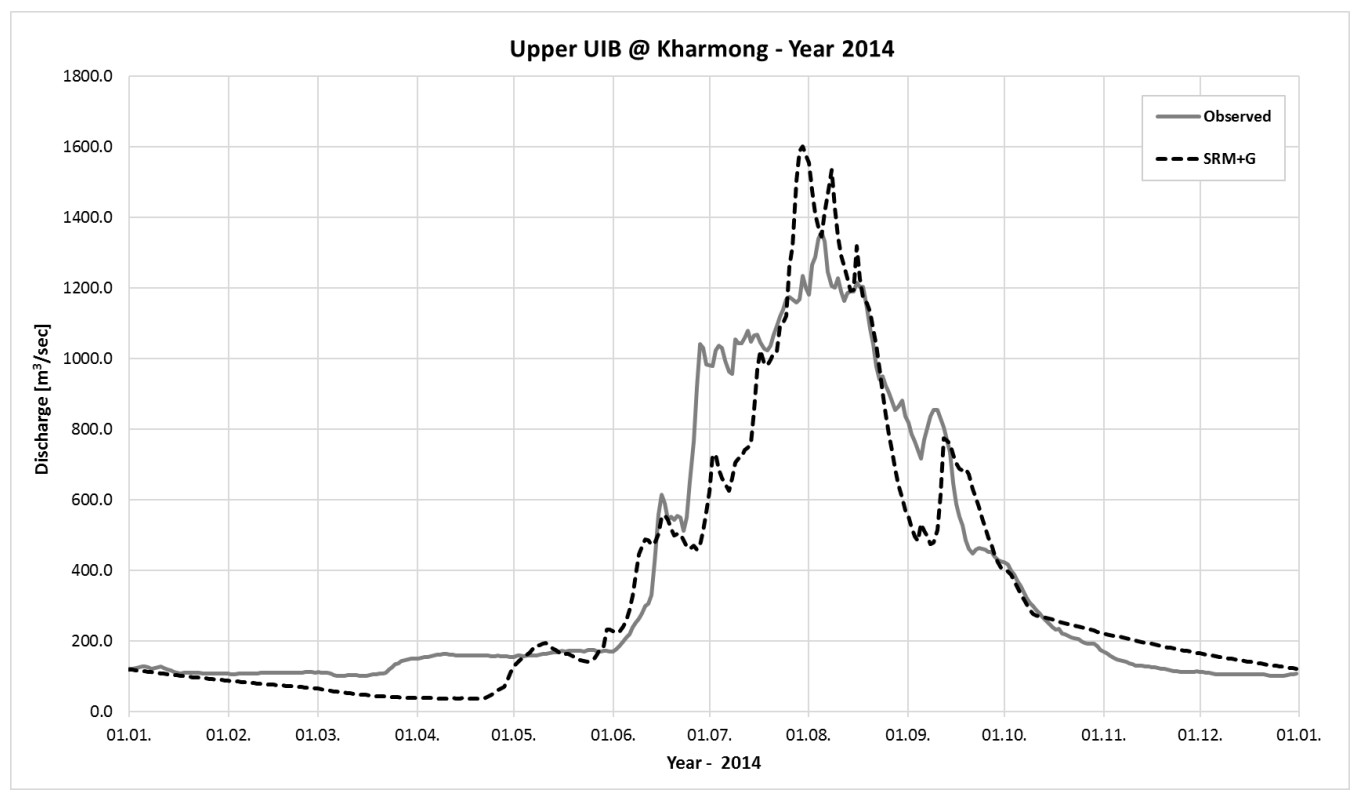

**Figure 10: Results of validation of final Upper UIB flow forecast model (dashed line) compared to observed flows at Kharmong (solid line) for the year 2014.**

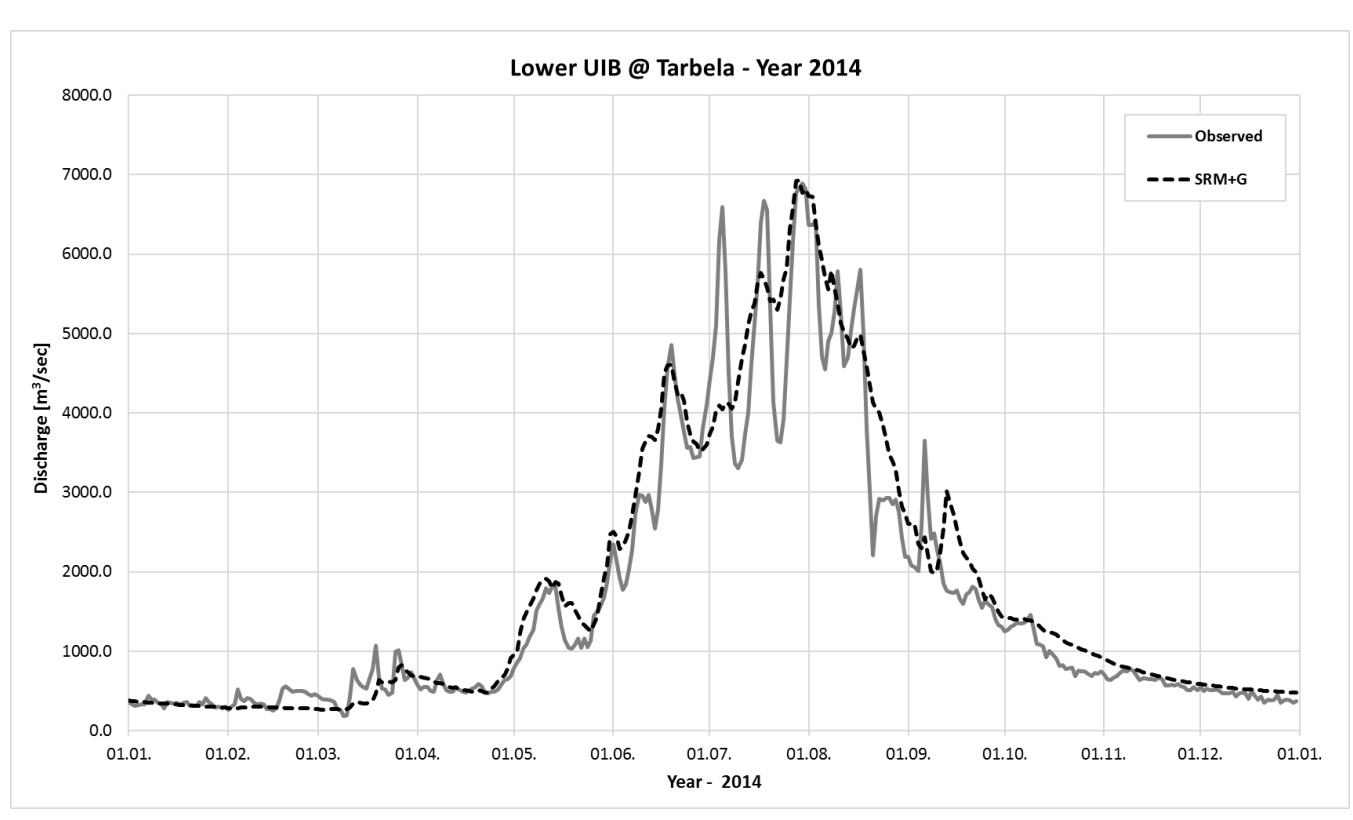

**Figure 11: Results of validation of final Lower UIB flow forecast model (dashed line) compared to observed inflows at Tarbela (solid line) for the year 2014.**

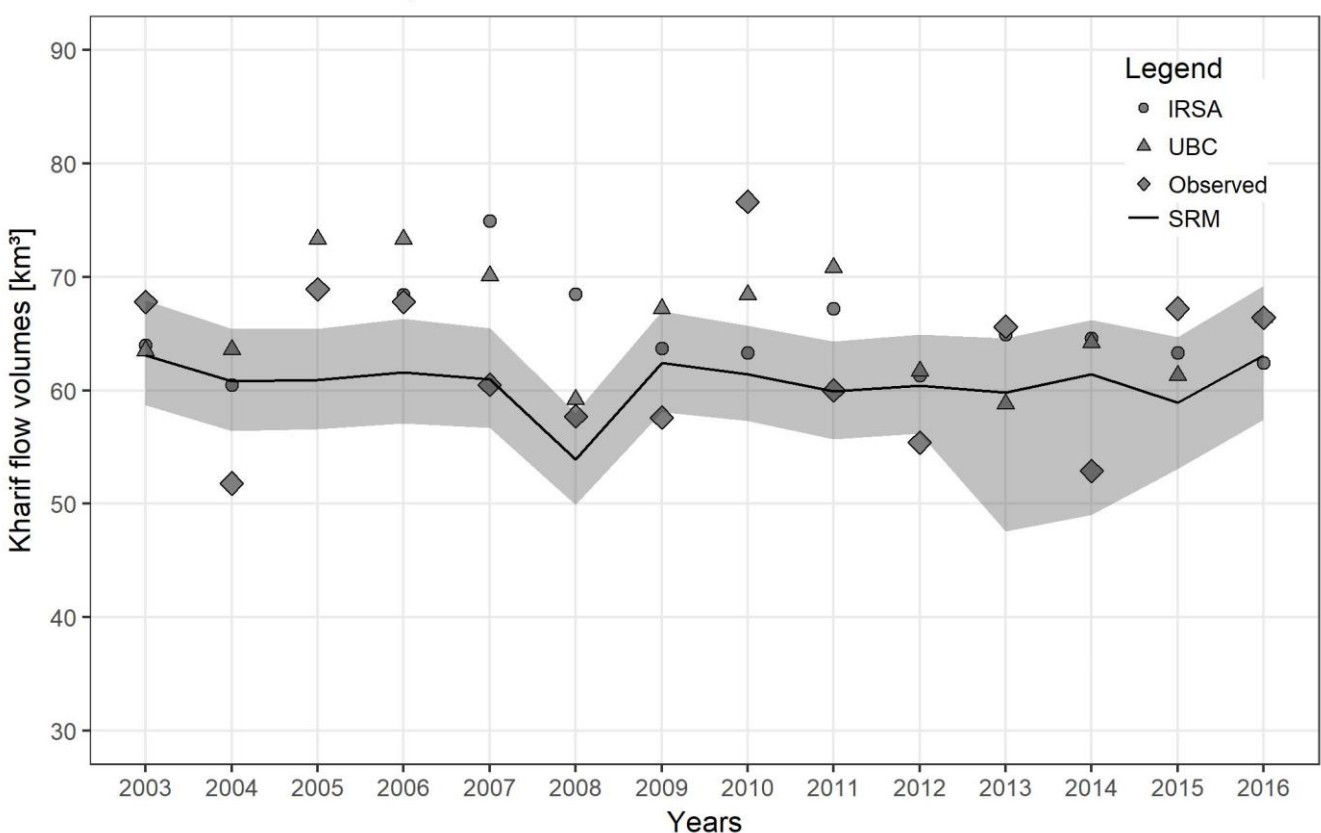

**Figure 12: Comparison of Kharif flow forecasts with 20% and 80% quantiles of SRM+G scenario ensembles years 2003 - 2016**

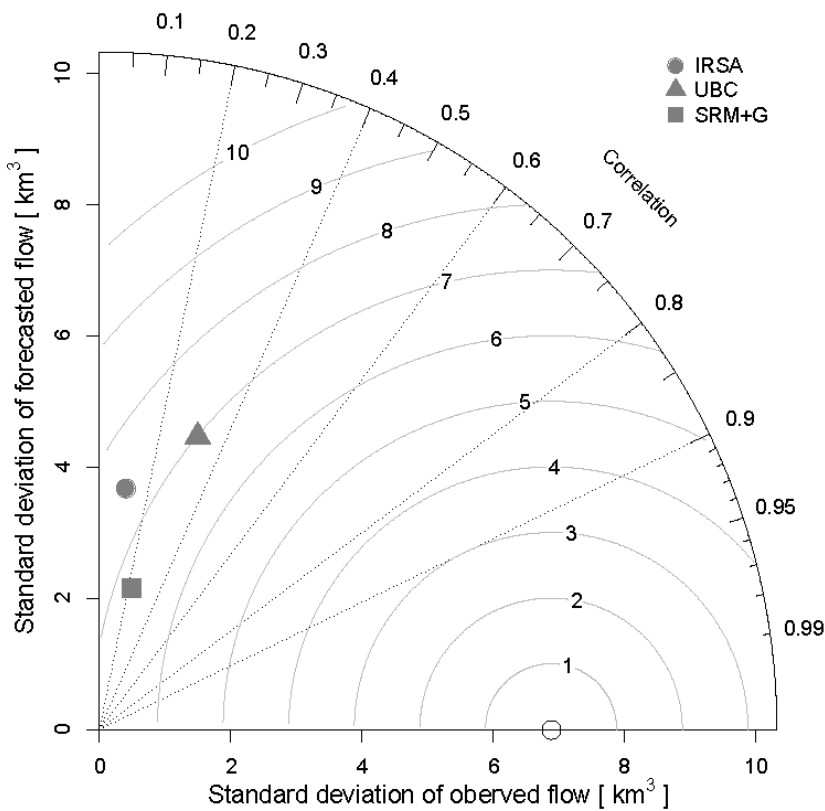

**Figure 13: Taylor diagram of IRSA, UBC, and SRM+G model performance**

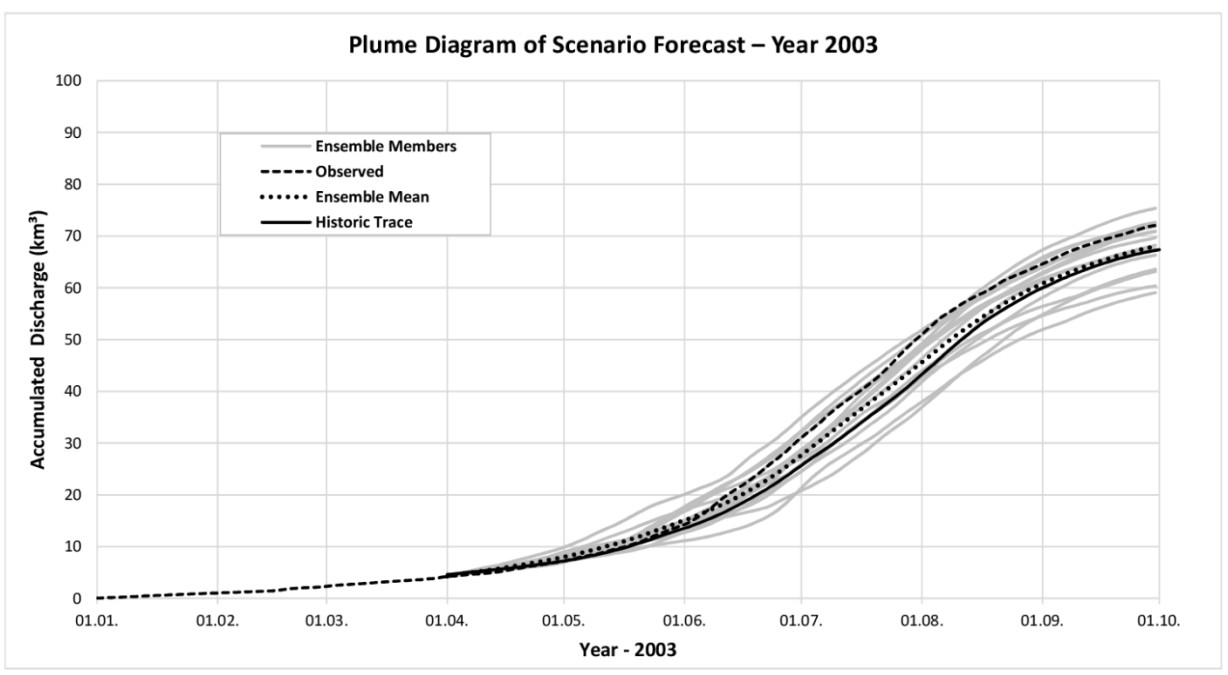

5  **Figure 14: Plume diagram of ensemble member traces in 2003**

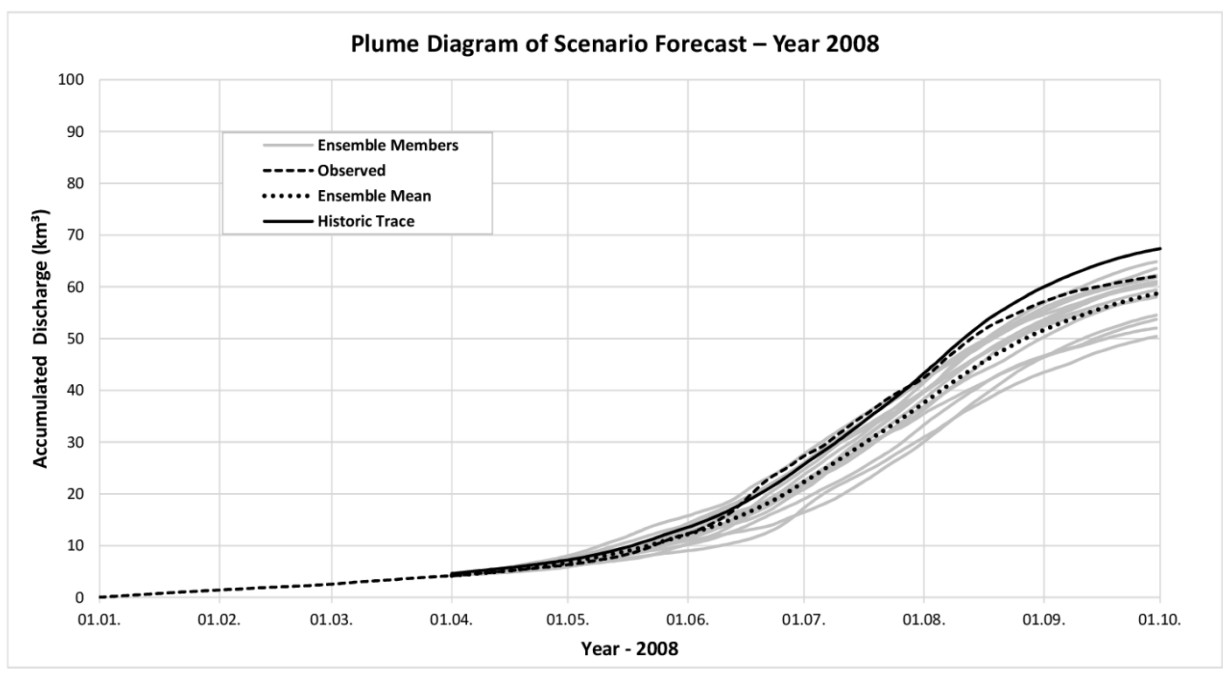

**Figure 15: Plume diagram of ensemble member traces in 2008**

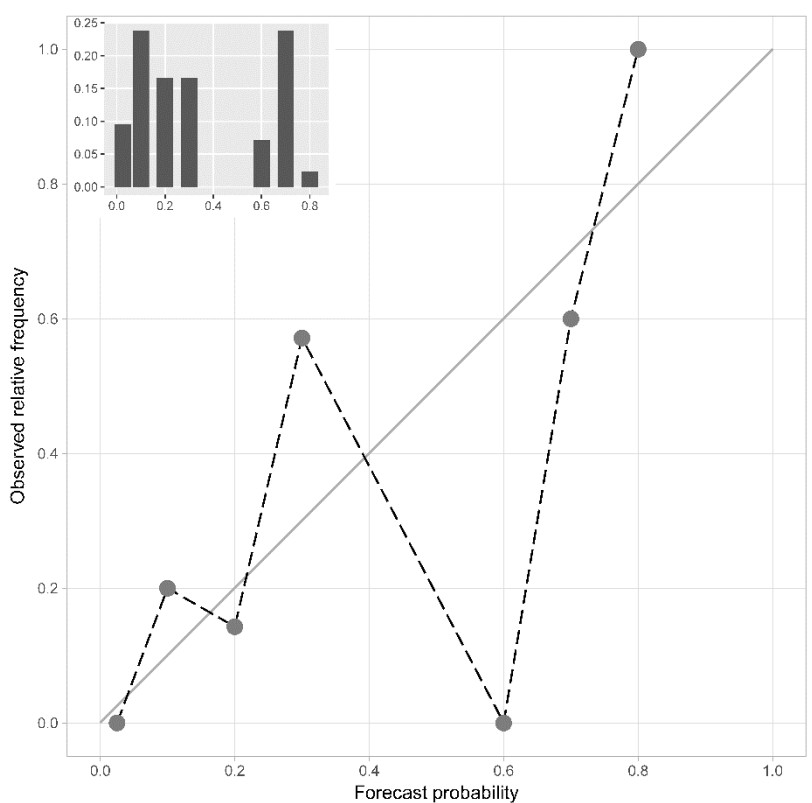

**Figure 16: Reliability diagram of probability forecasts in the categories dry (≤20%), near normal, and wet (>80%) of 2003 – 2014 flows**