# Peer review of "Scenario approach for the seasonal forecast of Kharif flows from Upper Indus Basin"

_Hydrology and Earth System Sciences, 2017_

## Referee Comment (RC1) · Anonymous Referee #1 · 15 May 2017

This manuscript presents the application of a Snowmelt & Runoff model, which has been extended to include a Glacier melt component, for the seasonal prediction of the Kharif season inflows to the Tarbela reservoir in the upper Indus basin. As the authors clearly point out, the accurate prediction of these inflows is of key importance to the planning of water allocation to the extensive downstream irrigation areas, which are vital to Pakistan's economy and food security. The development of such a seasonal inflow product is therefore very relevant. As the authors point out this is very challenging, given the complex topography as well as the scarcity of data.

Although the model and results presented in the manuscript are therefore certainly of interest, I am of the opinion that the manuscript is not yet of a maturity that is sufficient

for publication in HESS. I would, however, recommend and encourage the authors to develop this research further and improve the manuscript for resubmission to HESS or another Journal.

I have a number of concerns on the manuscript in its current form.

The embedding of the manuscript in the current state of the science of seasonal forecasting is poor. The authors do not review different methods that have been proposed or are in operational use for developing seasonal forecasts. I would expect some discussion on statistical methods as opposed to methods that use numerical weather prediction models. At the minimum the Ensemble Streamflow Prediction (ESP) approach, which has been widely used for seasonal water resources predictions should be considered. In fact, the scenario method that has been applied, where historical years are selected, which the authors present in Section 2.6 seems to have large similarities to the ESP. If the authors review some of the literature on the application of ESP they will understand that ESP is particularly skilful where the initial conditions are persistent, as is often the case where snowmelt dominates the water resource availability. Strangely, this approach is discussed in the methods section, but I could not identify in the results and discussion section the resulting ensembles or evaluation of the results of the scenarios selected. In assessing the skill of the forecasts I think the authors could significantly extend the current discussion and results presented. Perhaps the most interesting is the question is if the forecasts presented have improved skill when compared to one or more reference forecasts. While the errors the author report are appreciably small, I would wonder how these compare to the errors compared to those of climatological forecasts, or another reference forecast. A good climatological forecast could be obtained using the average snow extent and water equivalent for the basin as an initial state, rather than the current variable snow extent based on satellite data. A more elaborate verification of forecasts would be insightful. A start could be to see how the frequency of the monthly volumes created by the (ensemble) forecasts compare to the observed frequencies, as presented in Figure 2. This could

be developed further into a reliability diagram, or a rank-histogram, to name but a few commonly used statistical performance criteria commonly used to verify the skill of forecasting systems. I would recommend the authors consider developing a structured approach to these comparisons, where each model improvements proposed is rigorously evaluated against a benchmark to understand if that improvement is beneficial or detrimental to forecast skill.

The model the authors use is a very simple model, which in such a data scarce situation is a good choice in my opinion. However, despite that simplicity, there are some points that certainly deserve attention. In the way the model has been set up there appears to be a large set of parameters. It is not that clear in the manuscript how these have been derived during the calibration. Was this done as a pure calibration exercise, or was some physical basis used to estimate parameter values. This is perhaps the most apparent for the degree-day factors in Figure 6 and Table 2&3. First the unit should be added to the table, but assuming these are cm/day/C-1 then the values are in a reasonable range, if somewhat low at the start of the season. What is interesting is the linear relationship, with factors increasing during the season. I would like some more discussion on how realistic this is. Is there a physical reason that these melt-factors increase to such an extent, or is this the result of the calibration procedure? One reason for noting this is also that the hydrographs the authors present in figures 7&8 do not clearly show the contribution of the different sources of runoff that are considered in the model; direct runoff from precipitation; snowmelt; and glacier melt. As the authors note there are large uncertainties in the precipitation inputs. While RFE is chosen as an input, which given the scarcity of data is to my mind a reasonable choice, it is not so clear if there is an under-prediction of precipitation input. Comparison to the few ground stations may not help as these are, again as the authors note, likely underestimating the precipitation.

The simulated flood hydrographs are a combination of several inputs, including snowmelt and evolving monsoon. This could mean that higher degree day values required later on in the snowmelt season could be required in the calibration to compensate the underestimate of the precipitation input. This problem may be compounded due to the model presented not being mass conservative, and depends on the estimate (from satellite data) of the initial snow extent. The procedure that the authors present infers to my understanding the snow water equivalent through the selection of depletion curves. This is of course a novel approach given the lack of data, but it also contributes to model uncertainty. I think the manuscript can benefit from a more elaborate discussion/review of the methods proposed, as well as their plausibility in representing the different hydrological process that occur in the basin. This could include some thoughts on how the behaviour that is seen in some of the parameters, such as the increasing degree-day factors relate to the results of other research, or of observations. Although the model itself is not conservative, and does not include a base-flow component such as subsurface flow, the authors could evaluate the water balance across the fourteen year period they have selected. While this may include some assumption on glacier depletion over the period, it may be informative. Overall the presentation of the results and discussion is weak. Results of some of the methods described are not presented, such as the results of the ensemble scenario approach. Also, the hydrographs presented at the two gauging stations are for 2008, and seem not to represent real forecasts, but rather simulations. It is not clear. Forecasts have been developed for 2015 and 2016, but the results of these are not really presented, other than stating the estimated error. I would suggest the authors develop a much more structured approach to the results section; evaluating the model structure and its sensitivities, and then going on to explore the performance of the forecasts.

Finally, the manuscript lacks clear conclusions or take-home messages, as well as an outlook to scientific challenges that have been identified.

The Figures included are not clear and would need to be improved. Figure 3 is to a large extent redundant, and with the division of the basin easily displayed on Figure 1. The figures that include the hydrographs are also not easy to read.

---

## Referee Comment (RC2) · Anonymous Referee #2 · 20 May 2017

This paper introduced a scenario approach [actually the ensemble streamflow prediction (ESP)] to predict the Kharif flows from Upper Indus Basin (UIB) at seasonal timescale. Given the unique regional characteristics, the authors employed an improved Snowmelt Runoff Model (SRM) by incorporating the glacier component and divided the UIB region into Upper and Lower parts. The result shows this improved model (SRM+G)-based scenario approach seems to produce smaller overall mean absolute error in comparison with other different operational seasonal forecast models. As a regional case study, this study may be likely of interests of local researchers. However, I do not have a few major concerns.

Comments:

1. This paper is very short and looks like a letter. I am not sure whether it is suitable for HESS. Actually, some important information is missing (see my specific comments in following). 2. Section 2.2 (Line 11-14). You should provide enough quantitative evidences to explain how to divide the Upper and Lower parts within UIB. 3. Section 2.2 (Line 15-16). There are little details about the Kirpich travel-time equation that is used to determine the 3-day time lag between Kharmong and Tarbela. Please add the relevant information in the revised version. 4. Section 3 (Line 1-5). This part (forecast skill metrics) belongs to method description. Please move it to Section 2. 5. The authors only focused on the forecast performance of median (50%) volume values. Actually, the extreme volumes, like "dry" (20%) and "wet" (80%) conditions, may be of greater importance for the downstream regions. It should add a few additional skill assessments in terms of predicting extreme conditions. 6. This study assessed the forecast skill only by examining the volume difference of determined values. In the revised version, a few probabilistic quantitative metrics, like anomaly correlation (AC), Brier Score (BS), the false alarm rate (FAR), hit rate (HR), and Equitable Threat Score (ETS), should be employed to assess its skill in probabilistic forecasting. 7. Section 3. Please add 1-2 figures to illustrate the comparison of SRM+G model-based scenario approach with other forecast models. 8. You are suggested to add a comparison between SRM+G and SRM, to highlight the superiority of SRM+G in terms of incorporating glacier component. Also, a comparison of SRM+G estimates between with and without consideration of divisions (Upper and Lower parts) should be inserted and discussed in the revision. 9. Section 2.3 (Line 30). If I understand correctly, R indicates the daily runoff depth, not precipitation depth. 10. Section 2.5 (Line 33). "TRMM 3B34 product" should be "TRMM 3B34 product". Please correct it.

---

## Author Comment (AC1) · 24 Jun 2017

Dear Referee No. 2

Thank you for your specific comments and suggestion, we have tried to address your concerns on the points you have mentioned below. We appreciate your precise comments and detailed suggestion for the improvement of our paper.

COMMENT: Section 2.2 (Line 11-14). You should provide enough quantitative evidences to explain how to divide the Upper and Lower parts within UIB.

RESPONSE: The main reason for splitting the model into two parts is that the Modified

Depletion Curve approach for predicting the available snow-water equivalent, which strongly depends on a representative behavior of the elevation zones that are already melting at the time when the forecast is to be issued. In this respect, the snow cover at the Tibetan Plateau behaves very different from the remaining parts which leads to a severe underestimation of the actual snow-water equivalent. These differences in snow cover depletion are given in our comparison in Table 01 of the research paper. We will explain this in more detail also giving some figures which will exhibit this effect. Ideally the split between Upper and Lower UIB model should be done downstream of the Tibetan Plateau. Due to the lack of discharge data in this part of the UIB, Kharmong as the closest gauging station where daily discharge data is available was selected.

COMMENT: Section 2.2 (Line 15-16). There are little details about the Kirpich travel-time equation that is used to determine the 3-day time lag between Kharmong and Tarbela. Please add the relevant information in the revised version.

RESPONSE: We will give a short paragraph on the Kirpich formula as well as on the determination of the input values, i.e. channel flow length and main channel slope.

COMMENT: Section 3 (Line 1-5). This part (forecast skill metrics) belongs to method description. Please move it to Section 2.

RESPONSE: You are right. We shall move this to the methods section in the revised version.

COMMENT: The authors only focused on the forecast performance of median (50%) volume values. Actually, the extreme volumes, like "dry" (20%) and "wet" (80%) conditions, may be of greater importance for the downstream regions. It should add a few additional skill assessments in terms of predicting extreme conditions.

RESPONSE: We have confine ourselves to a comparison of the forecasted "most likely" (median) values, as only these are in hand for the IRSA and UBCWM forecasts. We shall add a figure giving the 20%, 50%, and 80% flows forecasted by SRM+G for all

years in comparison with the observed volumes, on which the forecast skill for extreme conditions will be discussed.

COMMENT: This study assessed the forecast skill only by examining the volume difference of determined values. In the revised version, a few probabilistic quantitative metrics, like anomaly correlation (AC), Brier Score (BS), the false alarm rate (FAR), hit rate (HR), and Equitable Threat Score (ETS), should be employed to assess its skill in probabilistic forecasting.

RESPONSE: We have focussed on the error in predicted flow volume, as this is the common metric in discussions with the involved authorities. We will discuss additional probabilistic metrics like the ACC in the revised version. We also may visualise these in a Tailor diagram etc. However it has to be considered, that forecasts by all of the three models are primarily based on estimates of the snow-water equivalent at the beginning of the Kharif season while the climatic conditions during the forecast period, including precipitation during monsoon, are basically averaged. Thus all models tend to predict near to average conditions.

COMMENT: Section 3. Please add 1-2 figures to illustrate the comparison of SRM+G model-based scenario approach with other forecast models.

RESPONSE: There are only two operational forecast models currently working in Pakistan. One is IRSA's statistical model and the other is the University of British Columbia Watershed Model (UBCWM) which is operated by the Glacier Monitoring and Research Centre (GMRC) of Water and Power Development Authority (WAPDA). In Table 04 of the research paper we have given a summarised comparison with these two models. Table 04 will be substituted by a more detailed comparison of yearly forecast results with all three models for the study period as given in the attached Table 01.

COMMENT: You are suggested to add a comparison between SRM+G and SRM, to highlight the superiority of SRM+G in terms of incorporating glacier component.

RESPONSE: We very much appreciate this suggestion as well as the one below, as this will more clearly point out the improvements obtained in the study. We have added a comparison of SRM (without Glaciers) and SRM+G (with Glaciers). In the attached Figure 01 one can see the impact of glaciers on the hydrograph while using the same set of parameters for both the models. The impact of glacier in the Lower UIB is much more prominent than in the Upper UIB. The reason is that in the Lower UIB the glaciated area is about 10.5% of the catchment area while in the Upper UIB the glaciated area is only 1.7%.

COMMENT: Adding a comparison of SRM+G estimates between with and without consideration of divisions (Upper and Lower parts) should be inserted and discussed in the revision.

RESPONSE: We will give a comparison of forecast results and a more detailed discussion on this. Please see also our response to the 1st comment.

COMMENT: Section 2.3 (Line 30). If I understand correctly, R indicates the daily runoff depth, not precipitation depth.

RESPONSE: Yes, you are right, precipitation depth relates to the original SRM notation. We shall re-write this in the revised version.

COMMENT: Section 2.5 (Line 33). "TRMM 3B34 product" should be "TRMM 3B34 product". Please correct it.

RESPONSE: We have written "TRMM 3B34 product" in our paper. So please let us know what your point is.

Please also note the supplement to this comment:
http://www.hydrol-earth-syst-sci-discuss.net/hess-2017-182/hess-2017-182-AC1-supplement.pdf
* * *
[Figure]

182, 2017.

Table 01 : Yearly comparison of forecasted Kharif flows [10⁶m³] for the three operational forecast models

| Year | Observed | ISA | | | FSM+G | | | USGVM | | |
|---|---|---|---|---|---|---|---|---|---|---|
| | | Most Likely | Error | \|Error\| | Most Likely | Error | \|Error\| | Most Likely | Error | \|Error\| |
| 2003 | 67775.7 | 63960.6 | -5.6% | 5.6% | 63099.6 | -7.0% | 7.0% | 63468.6 | -6.0% | 6.0% |
| 2004 | 51783.5 | 60516.6 | 16.9% | 16.9% | 60762.6 | 17.0% | 17.0% | 63591.6 | 23.0% | 23.0% |
| 2005 | 68880.7 | 69008.7 | 0.2% | 0.2% | 60885.6 | -12.0% | 12.0% | 73308.7 | 6.0% | 6.0% |
| 2006 | 67775.7 | 68388.7 | 0.9% | 0.9% | 61623.6 | -9.0% | 9.0% | 73308.7 | 8.0% | 8.0% |
| 2007 | 60516.6 | 74907.7 | 23.8% | 23.8% | 61008.6 | 1.0% | 1.0% | 70110.7 | 16.0% | 16.0% |
| 2008 | 57687.6 | 68511.7 | 18.8% | 18.8% | 53874.5 | -7.0% | 7.0% | 59163.6 | 3.0% | 3.0% |
| 2009 | 57564.6 | 63714.6 | 10.7% | 10.7% | 62361.6 | 8.0% | 8.0% | 67158.7 | 17.0% | 17.0% |
| 2010 | 76629.8 | 63345.6 | -17.3% | 17.3% | 61377.6 | -20.0% | 20.0% | 68388.7 | -11.0% | 11.0% |
| 2011 | 60024.6 | 67158.7 | 11.9% | 11.9% | 59901.6 | 0.0% | 0.0% | 70848.7 | 18.0% | 18.0% |
| 2012 | 55350.6 | 61254.6 | 10.7% | 10.7% | 60393.6 | 9.0% | 9.0% | 61746.6 | 12.0% | 12.0% |
| 2013 | 65559.7 | 64944.6 | -0.9% | 0.9% | 59778.6 | -9.0% | 9.0% | 58794.6 | -10.0% | 10.0% |
| 2014 | 52890.5 | 64575.6 | 22.1% | 22.1% | 61377.6 | 16.0% | 16.0% | 64206.6 | 21.0% | 21.0% |
| 2015 | 67158.7 | 63345.6 | -5.7% | 5.7% | 58917.6 | -12.0% | 12.0% | 61254.6 | -9.0% | 9.0% |
| 2016 | 66420.7 | 62361.6 | -6.1% | 6.1% | 63062.7 | -5.0% | 5.0% | 66420.7 | 0.0% | 0.0% |
| Bias / Mean Absolute Error | | | 5.7% | 10.6% | | -2.1% | 9.4% | | 6.3% | 11.4% |

Fig. 1.
* * *
Interactive
comment

[Figure]

Figure 01: Comparison of model with and without glaciers

**Fig. 2.**

---

## Author Comment (AC2) · 28 Jun 2017

Dear Referee No. 1

We thank you very much for your valuable comments. These will definitely improve our paper. We have tried to answer your main remarks by splitting them into separate comments, which as we hope give a correct interpretation of your major points.

COMMENT: The authors do not review different methods that have been proposed or are in operational use for developing seasonal forecasts. I would expect some discussion on statistical methods as opposed to methods that use numerical weather

prediction models. At the minimum the Ensemble Streamflow Prediction (ESP) approach, which has been widely used for seasonal water resources predictions should be considered.

RESPONSE: We agree to your comment. We have focussed on the enhancement of SRM+G (mainly inclusion of glacier melt and splitting of the UIB into sub-catchments) in comparison to the other operational forecast models used in Pakistan and not so much on the scenario approach that we have applied. In fact, this approach is quite similar to the NWS Extended Streamflow Prediction / Ensemble Streamflow Prediction (ESP) approach. However it is rather aimed at arriving at a point forecast than to explicitly generate a probabilistic forecast. We will include a discussion on this topic in a revised version.

COMMENT: I could not identify in the results and discussion section the resulting ensembles or evaluation of the results of the scenarios selected. In assessing the skill of the forecasts I think the authors could significantly extend the current discussion and results presented. (. . .) A more elaborate verification of forecasts would be insightful.

RESPONSE: As stated in our response to a similar comment by Referee No. 2, we have focussed on the error in predicted flow volume, as this is the common metric in discussions with the involved authorities. We will substitute Table 4 of the paper by a more detailed comparison of yearly forecast results from all three models as given in the attached Table 01. Furthermore, as suggested by both referees, the forecast skills will be evaluated against reference (climatological) forecasts by quantitative metrics and respective diagrams used in probabilistic forecast verification.

COMMENT: It is not that clear in the manuscript how these [model parameters] have been derived during the calibration. Was this done as a pure calibration exercise, or was some physical basis used to estimate parameter values?

RESPONSE: Apart from the degree-day factors (see next comment) the model parameters have been derived mainly as described in the SRM user manual (Martinec et al.,

2011). Some were slightly adjusted by manual fitting. We may show a synopsis of the model parameters as given in attached Table 02 and explain briefly how each of them was derived.

COMMENT: What is interesting is the linear relationship, with [degree-day] factors increasing during the season. I would like some more discussion on how realistic this is. Is there a physical reason that these melt-factors increase to such an extent, or is this the result of the calibration procedure?

RESPONSE: Many researcher have addressed the temporal and spatial variability of degree-day factors, e.g. Hock (2003), van den Broeke et al. (2010), and others. However, the complex interactions between atmospheric and surface characteristics affecting the degree-day factor is still not very well understood (He et al., 2014). One mechanism obviously is the accumulation of energy from solar radiation as well as from air temperature during the 'ripening' of the snowpack that is different with altitude. The authors have shown and discussed the temporal increase in a paper by Ismail et al. (2015). We may give a more detailed explanation in a revised version.

COMMENT: While RFE is chosen as an input, which given the scarcity of data is to my mind a reasonable choice, it is not so clear if there is an under-prediction of precipitation input. Comparison to the few ground stations may not help as these are, again as the authors note, likely underestimating the precipitation.

RESPONSE: We fully agree that, as others also have noted, ground stations due to their location at the valley floors tend to underestimate the actual precipitation. We therefore have referred in section 2.5 to a paper by Reggiani and Rientjnes (2015) who have compared a number of different studies with own calculations and estimate the mean annual precipitation in UIB to 675 ±100 mm/a, which is higher than in most of the other studies. The RFE basin-wide annual mean for the period 2003 – 2015 is 701 mm/a, which corresponds to the ERA-Interim (681 mm/a) and NCEP/NCAR (705 mm/a) reanalysis means calculated by Reggiani and Rientjnes. Taking into account

the much lower values of other precipitation (reanalysis) products as well as of most of the UIB water balance studies, we feel that the RFE product is more likely over- rather than under-estimating the actual precipitation.

COMMENT: . . . the hydrographs the authors present in figures 7&8 do not clearly show the contribution of the different sources of runoff that are considered in the model; direct runoff from precipitation; snowmelt; and glacier melt. (. . .) Although the model itself is not conservative, and does not include a base-flow component such as subsurface flow, the authors could evaluate the water balance across the fourteen-year period they have selected.

RESPONSE: It is not possible to split the calculated daily discharge into the different sources of runoff, as due to SRM's recession flow approach (eq. 1 in the paper) only a small part (about 10%) contributes directly to the daily discharge while the larger part origins from recession flow $Q_n$. In this respect the model is not mass-conservative. Annual water balances give an average contribution of 26%, 53%, and 21% from rain, snowmelt, and glacier melt respectively, which coincides well with figures given by Immerzeel et al. (2010) or Charles (2016).

COMMENT: Overall the presentation of the results and discussion is weak. Results of some of the methods described are not presented, such as the results of the ensemble scenario approach. Also, the hydrographs presented at the two gauging stations are for 2008, and seem not to represent real forecasts, but rather simulations. (. . .) Forecasts have been developed for 2015 and 2016, but the results of these are not really presented, other than stating the estimated error.

RESPONSE: As stated before, we will substitute Table 4 of the paper by a yearly comparison of forecast results with all three models as given in the attached Table 01. Regarding SRM+G, figures for the years 2003 – 2014 are from hindcasts using the developed forecasting procedures, while 2015 and 2016 are genuine forecasts. Fig. 7 & 8 indeed show the results of a model simulation aimed to validate the forecasting

parameter sets and rules (= historical trace). In addition we shall give e.g. a plume diagram showing the traces of all members of the forecast ensemble.

COMMENT: Finally, the manuscript lacks clear conclusions or take-home messages, as well as an outlook to scientific challenges that have been identified.

RESPONSE: The authors think that they gave some conclusions, e.g. on improving the UIB SRM+G model (inclusion of glacier melt, splitting of the catchment), as well as an outlook on further fields of research, e.g. selection of scenarios respectively weighting of ensemble members according to climate signals. Nevertheless we appreciate the comment as it is indicating, that this is not communicated clearly and we will give a separate and extended chapter 'Conclusions' in a revised version.

COMMENT: The Figures included are not clear and would need to be improved. Figure 3 is to a large extent redundant, and with the division of the basin easily displayed on Figure 1. The figures that include the hydrographs are also not easy to read.

RESPONSE: We have included the splitting of UIB (Figure 3) in the catchment map as suggested (see attached Figure 1) and have simplified the hydrographs.

REFERENCES:

Charles, S.P. (2016). Hydroclimate of the Indus ‐ synthesis of the literature relevant to Indus basin hydroclimate processes, trends, seasonal forecasting and climate change. CSIRO Sustainable Development Investment Portfolio project. CSIRO Land and Water, Australia.

He, Z. H.; Parajka, J.; Tian, F. Q.; Blöschl, G. (2014): Estimating degree-day factors from MODIS for snowmelt runoff modeling. In Hydrol. Earth Syst. Sci. 18 (12), 4773–4789.

Hock, R., (2003): Temperature index melt modelling in mountain areas. Journal of Hydrology 282 (1-4), 104–115.

Immerzeel, W.W.; Droogers, P.; de Jong, S.M.; Bierkens, M.F.P. (2009). Satellite Derived Snow and Runoff Dynamics in the Upper Indus River Basin. Grazer Schriften der Geographie und Raumforschung, Volume 45, 303 - 312

Ismail, M. F., Rehman, H., Bogacki, W., and Noor, M. (2015): Degree Day Factor Models for Forecasting the Snowmelt Runoff for Naran Watershed, Sci. Int. Lahore, 27, 1961–1969. https://www.researchgate.net/publication/310064814_Degree_Day_Factor_Models_for_Forecasting_the_Snowmelt_Runo

Martinec, J., Rango, A., and Roberts, R. (2011): Snowmelt Runoff Model User's Manual, WinSRM Version 1.14. Agricultural Experiment Station Special Report 100, New Mexico State University, Las Cruces, NM 88003, USA, 2011.

Reggiani, P. and Rientjes, T. H. M. (2015). A reflection on the long-term water balance of the Upper Indus Basin. Hydrol. Res., 46(3), 446-462.

van den Broeke, M., C. Bus, J. Ettema, and P. Smeets (2010). Temperature thresholds for degree-day modelling of Greenland ice sheet melt rates. In Geophys. Res. Lett. 37 (18),

Please also note the supplement to this comment:
https://www.hydrol-earth-syst-sci-discuss.net/hess-2017-182/hess-2017-182-AC2-supplement.pdf

Table 01: Yearly comparison of forecasted Kharif flows [$10^6$ m$^3$] for the three operational forecast models

| Year | Observed | IRSA Most Likely | Error | \|Error\| | SRM+G Most Likely | Error | \|Error\| | UBCWM Most Likely | Error | \|Error\| |
|------|----------|-------------------|-------|-----------|--------------------|-------|-----------|---------------------|-------|-----------|
| 2003 | 67773.7 | 63960.6 | -5.6% | 5.6% | 63099.6 | -7.0% | 7.0% | 63468.6 | -6.0% | 6.0% |
| 2004 | 51783.5 | 60516.6 | 16.9% | 16.9% | 60762.6 | 17.0% | 17.0% | 63591.6 | 23.0% | 23.0% |
| 2005 | 68880.7 | 69003.7 | 0.2% | 0.2% | 60885.6 | -12.0% | 12.0% | 73308.7 | 6.0% | 6.0% |
| 2006 | 67773.7 | 68388.7 | 0.9% | 0.9% | 61623.6 | -9.0% | 9.0% | 73308.7 | 8.0% | 8.0% |
| 2007 | 60516.6 | 74907.7 | 23.8% | 23.8% | 61008.6 | 1.0% | 1.0% | 70110.7 | 16.0% | 16.0% |
| 2008 | 57687.6 | 68511.7 | 18.8% | 18.8% | 53874.5 | -7.0% | 7.0% | 59163.6 | 3.0% | 3.0% |
| 2009 | 57564.6 | 63714.6 | 10.7% | 10.7% | 62361.6 | 8.0% | 8.0% | 67158.7 | 17.0% | 17.0% |
| 2010 | 76629.8 | 63345.6 | -17.3% | 17.3% | 61377.6 | -20.0% | 20.0% | 68388.7 | -11.0% | 11.0% |
| 2011 | 60024.6 | 67158.7 | 11.9% | 11.9% | 59901.6 | 0.0% | 0.0% | 70848.7 | 18.0% | 18.0% |
| 2012 | 55350.6 | 61254.6 | 10.7% | 10.7% | 60393.6 | 9.0% | 9.0% | 61746.6 | 12.0% | 12.0% |
| 2013 | 65559.7 | 64944.6 | -0.9% | 0.9% | 59778.6 | -9.0% | 9.0% | 58794.6 | -10.0% | 10.0% |
| 2014 | 52890.5 | 64575.6 | 22.1% | 22.1% | 61377.6 | 16.0% | 16.0% | 64206.6 | 21.0% | 21.0% |
| 2015 | 67158.7 | 63345.6 | -5.7% | 5.7% | 58917.6 | -12.0% | 12.0% | 61254.6 | -9.0% | 9.0% |
| 2016 | 66420.7 | 62361.6 | -6.1% | 6.1% | 63062.7 | -5.0% | 5.0% | 66420.7 | 0.0% | 0.0% |
| Bias / Mean Absolute Error | | | 5.7% | 10.8% | | -2.1% | 9.4% | | 6.3% | 11.4% |

Fig. 1.

Table 02:  SRM+G Model Parameters

| Parameter | Symbol | Value | Units | Remarks |
|---|---|---|---|---|
| Temperature Lapse-Rate | $\gamma$ | 6.0 | °C km$^{-1}$ | |
| Recession Coefficient | $k_x$ | 1.193 | – | October-February |
| | | 1.060 | – | March – September |
| | $k_y$ | 0.029 | | October-February |
| | | 0.020 | | March – September |
| Critical Precipitation | $P_{crit}$ | 1 | cm | |
| Lag Time | $L$ | 54 | h | 2.5 days delay between melt and runoff at Tarbela |
| Critical Temperature | $T_{crit}$ | 0.5 – 3.0 | °C | |
| Rainfall Contributing Area | $RCA$ | 0 | – | November – March |
| | | 1 | | April – October |
| Runoff Coefficient Snow | $c_S$ | 0.8 | – | |
| Runoff Coefficient Glacier | $c_G$ | 0.7 | – | |
| Runoff Coefficient Rain | $c_R$ | 0.40 – 0.75 | – | |
| Degree-Day Factor Snow | $\alpha$ | 0.15 – 0.80 | cm °C$^{-1}$d$^{-1}$ | zone-wise and temporal varying (see Table 2 & 3) |
| Degree-Day Factor Glacier | $a_G$ | 0.7 | cm °C$^{-1}$d$^{-1}$ | |

**Fig. 2.**

Fig. 3.

Background Map Source: ESRI Open Street Map

**Upper UIB @ Kharmong - 2008**

Observed
Simulated

**Fig. 4.**

**Lower UIB @ Tarbela - 2008**

Observed

Simulated

**Fig. 5.**

---

## Author Response (AR1)

**Scenario approach for the seasonal forecast of Kharif flows from Upper Indus Basin**

Muhammad Fraz Ismail and Wolfgang Bogacki

Department of Architectural and Civil Engineering, Koblenz University of Applied Sciences, Germany.

*Correspondence to*:  W. Bogacki (bogacki@hs-koblenz.de)

**Abstract.** Snow and glacial melt runoff are the major sources of water contribution from the high mountainous terrain of Indus river upstream of the Tarbela reservoir. A reliable forecast of seasonal water availability for the Kharif cropping season (April – September) can pave the way towards the better water management and subsequently boost the agro-economy of Pakistan. The use of degree-day models in conjunction with the satellite based remote sensing data for the forecasting of seasonal snow and ice melt runoff has proved to be a suitable approach for the data scarce regions. In the present research, Snowmelt Runoff Model (SRM) has not only been enhanced by incorporating the "glacier (G)" component but also applied for the forecast of seasonal water availability from the Upper Indus Basin (UIB). Excel based SRM+G takes into account of separate degree-day factors for snow and glacier melt processes. All year simulation runs with SRM+G for the period 2003 – 2014 result in an average flow component distribution of 53%, 21%, and 26% for snow, glacier and rain respectively. The UIB has been divided into Upper and Lower parts because of the different climatic conditions in the Tibetan plateau. The scenario approach is a step towards probabilistic forecasting of seasonal flows in the UIB. As the accuracy of existing forecasts with a mean volume error of 10.9% and 11.4% is already quite high, the improvement by SRM+G having a MAPE of 9.5% is only limited. The bias however could be reduced to -2.0%. The challenge of course is to forecast the seasonal anomaly in temperature and precipitation. In this respect further research is needed on how today's global forecast systems may allow a more specific selection of ensemble members particularly in the UIB, where the correlation to common teleconnections like the ENSO status is known to be weak. The application of seasonal scenario based approach proved to be very adequate for long term water availability forecast.

**1 Introduction**

Mountains are the water towers of the world. They are the biggest resource of freshwater to half of the world's population fulfilling their needs for irrigation, industry, domestic and hydropower (Viviroli et al., 2007). The Indus River on which Pakistan's socio-economic development depends can be termed as the bread basket of Pakistan

5  (Clarke, 2015). Due to agrarian economy, Pakistan's agriculture share in water usage is about 97%, which is well above the global average of about 70% (Akram, 2009). In Pakistan, Indus River System Authority (IRSA) decides the provincial water shares according to the Water Apportionment Accord (WAA) of 1991 and provincial irrigation departments subsequently determine the seasonal water allocation to the different canal command areas depending upon the water availability forecast carried out at the end of March for the forthcoming Kharif cropping season

10  (April-September). A reliable seasonal forecast of the water availability from snow and glacial melt is therefore of utmost importance for the agricultural production and efficient water management.

But on the other hand snowmelt runoff modelling in mountainous regions faces the challenge of data scarcity as well as the uncertainty in parameter calibration (Pellicciotti et al., 2012). The need of the hour is to not only develop such a hydrological model which has the capability to cater both snow and glacial melt component but also a

15  reliable forecast technique which could help the water managers and policy makers for enhancing the water resources management in future. Present paper focuses on the implementation of snow and glacial melt runoff model on Upper Indus Basin (UIB) which in principle is fed by snow and glacial melt component 
[revised manuscript text omitted]

---

## Author Response (AR2)

**Point by point response to the comments**

**Scenario approach for the seasonal forecast of Kharif flows from Upper Indus Basin**

**Muhammad Fraz Ismail and Wolfgang Bogacki**

bogacki@hs-koblenz.de

Dear Referee No. 1

We thank you very much for your valuable comments. These will definitely improve our paper. We have tried to answer your main remarks by splitting them into separate comments, which as we hope give a correct interpretation of your major points.

**COMMENT:** The authors do not review different methods that have been proposed or are in operational use for developing seasonal forecasts. I would expect some discussion on statistical methods as opposed to methods that use numerical weather prediction models. At the minimum the Ensemble Streamflow Prediction (ESP) approach, which has been widely used for seasonal water resources predictions should be considered.

**RESPONSE**: In the revised manuscript in section *2.6 Scenario Approach of forecasting,* we have discussed the differences and similarities of Ensemble Stream Flow Prediction (ESP) method and the scenario approach of forecasting.

**COMMENT:** I could not identify in the results and discussion section the resulting ensembles or evaluation of the results of the scenarios selected. In assessing the skill of the forecasts I think the authors could significantly extend the current discussion and results presented. (…) A more elaborate verification of forecasts would be insightful.

**RESPONSE:**

In the new section *3.4 Evaluation of forecast skills,* we have included a more detailed discussion on the forecast skills and give a yearly comparison of the forecasts (Table 6) as well some verification metrics (Table 7). We also have included examples of plume diagrams of ensemble member trace have been given in the Figure 14 and 15 in the revised manuscript.

**COMMENT:** It is not that clear in the manuscript how these [model parameters] have been derived during the calibration. Was this done as a pure calibration exercise, or was some physical basis used to estimate parameter values?

**RESPONSE**: In section *2.5 Model parameters* are explained as well as in Table 3 of the revised manuscript all the parameters and their ranges are given.

**Table 3: SRM+G Model Parameters for both Upper and Lower UIB**

| Parameters | Symbol | Value | Units | Remarks |
|---|---|---|---|---|
| Temperature Lapse-Rate | $\gamma$ | 6.0 | °C km$^{-1}$ | |
| Recession Coefficient | $k_x$ | 1.193 | | October-February |
| | | 1.060 | | March – September |
| | $k_y$ | 0.029 | – | October-February |
| | | 0.020 | | March – September |

| Parameters | Symbol | Value | Units | Remarks |
|---|---|---|---|---|
| Critical Precipitation | $P_{crit}$ | 1 | cm | constant |
| Lag Time | $L$ | 54 | h | 2.5 days delay between melt and runoff at Tarbela |
| Critical Temperature | $T_{crit}$ | 0.5 – 3.0 | °C | variable |
| Rainfall Contributing Area | $RCA$ | 0
1 | – | November – March
April – October |
| Runoff Coefficient - Snow | $c_S$ | 0.80 | – | constant |
| Runoff Coefficient - Glacier | $c_G$ | 0.70 | – | constant |
| Runoff Coefficient - Rain | $c_R$ | 0.25-0.75 | – | |
| Degree-Day Factor - Snow | $\alpha$ | 0.15-0.80 | cm °C$^{-1}$d$^{-1}$ | |
| Degree-Day Factor - Glacier | $a_G$ | 0.70 | cm °C$^{-1}$d$^{-1}$ | constant |

**COMMENT:** What is interesting is the linear relationship, with [degree-day] factors increasing during the season. I would like some more discussion on how realistic this is. Is there a physical reason that these melt-factors increase to such an extent, or is this the result of the calibration procedure?

**RESPONSE**: A detailed discussion for the linear increase of the degree-day factors is given in section *2.5 Model parameters*. To summarise, the increase of the degree-day factors with the passage of time is because the snow absorbs energy due to its physical condition, in terms of increasing temperatures and solar radiations intensities. This process of energy storage plays a pivotal role in the ripening of the snowpack, which melts rapidly as the snow melting season progresses. The extent to which degree-day factors increase is related to the calibration procedure because it was observed during the model calibration that in a certain elevation zone when the degree-day factors attain the value e.g (0.80 cm °C$^{-1}$ d$^{-1}$), the snow cover area in that very elevation zone has almost completely faded away so there is no need to further increase the values of degree-day factors. The limit to what extent the degree-day factors increase at a certain spatio-temporal region depends upon various physiographic and climatic parameters and a research is on-going for evaluating the trend of degree-day factors in response to the aforementioned parameters.

**COMMENT:** While RFE is chosen as an input, which given the scarcity of data is to my mind a reasonable choice, it is not so clear if there is an under-prediction of precipitation input. Comparison to the few ground stations may not help as these are, again as the authors note, likely underestimating the precipitation.

**RESPONSE**: We fully agree that, as others also have noted, ground stations due to their location at the valley floors tend to underestimate the actual precipitation. We therefore have referred in section *2.4 Data sources* to a paper by Reggiani and Rientjnes (2015) who have compared a number of different studies with own calculations and estimate the mean annual precipitation in UIB to 675 ±100 mm/a, which is higher than in most of the other studies. The RFE basin-wide annual mean for the period 2003 – 2015 is 701 mm/a, which corresponds to the ERA-Interim (681 mm/a) and NCEP/NCAR (705 mm/a) reanalysis means calculated by Reggiani and Rientjnes.

**COMMENT:** The hydrographs, the authors present in figures 7&8 do not clearly show the contribution of the different sources of runoff that are considered in the model; direct runoff from

precipitation; snowmelt; and glacier melt. (…) Although the model itself is not conservative, and does not include a base-flow component such as subsurface flow, the authors could evaluate the water balance across the fourteen-year period they have selected.

**RESPONSE**: It is not possible to split the calculated daily discharge into the different sources of runoff, as due to SRM's recession flow approach (eq. 1 in the revised manuscript) only a small part (about 10%) contributes directly to the daily discharge while the larger part origins from recession flow $Q_n$. In this respect the model is not mass-conservative. Annual water balances give an average contribution of 26%, 53%, and 21% from rain, snowmelt, and glacier melt respectively, which coincides well with figures given by Immerzeel et al. (2010) or Charles et al. (2017).

However, we have now included the Figure 7 in the revised manuscript to show the monthly distribution of flow component in the UIB from 2003 - 2014. We have also updated the Figures 10 and 11 in order to make them more elaborative by showing the validation of SRM+G.

[Figure]

[Figure]

[Figure]

**Figure 7: Monthly distribution of flow components in UIB**

[Figure]

**Figure 10: Results of validation of final Upper UIB flow forecast model (dashed line) compared to observed flows at Kharmong (solid line) for the year 2014.**

[Figure]

**Figure 11: Results of validation of final Lower UIB flow forecast model (dashed line) compared to observed inflows at Tarbela (solid line) for the year 2014.**

**COMMENT:** Overall the presentation of the results and discussion is weak. Results of some of the methods described are not presented, such as the results of the ensemble scenario approach. Also, the hydrographs presented at the two gauging stations are for 2008, and seem not to represent real forecasts, but rather simulations. (…) Forecasts have been developed for 2015 and 2016, but the results of these are not really presented, other than stating the estimated error.

**RESPONSE**: We have tried to improve the results and discussion by separating it into different parts including, the splitting of the UIB, incorporation of glacier component in the SRM model, simulation model verification and evaluation of forecasting skills.

We have also substituted Table 6 in the paper providing a yearly comparison of forecast results with all three models. Regarding SRM+G, figures for the years 2003 – 2014 are from hind-casts using the developed forecasting procedures, while 2015 and 2016 are genuine forecasts. Figures 10 and 11 (see previous response) indeed show the results of a model simulation aimed to validate the forecasting parameter sets and rules

**COMMENT:** The Figures included are not clear and would need to be improved. Figure 3 is to a large extent redundant, and with the division of the basin easily displayed on Figure 1. The figures that include the hydrographs are also not easy to read.

**RESPONSE**: We have combined the figure 1 and figure 3 of the old manuscript into one figure as shown in Figure 1 of the revised manuscript. This Figure 1 shows the UIB boundary with different elevation zones as well as the catchment split line.

[Figure]

**Figure 1: Map of the Upper Indus Basin showing different elevations and splitting of UIB at the Kharmong gauging station into Upper and Lower UIB**

**COMMENT:** Finally, the manuscript lacks clear conclusions or take-home messages, as well as an outlook to scientific challenges that have been identified.

**RESPONSE**: The authors think that they have given some conclusions, e.g. on improving the UIB SRM+G model (inclusion of glacier melt, splitting of the catchment), as well as an outlook on further fields of research, e.g. selection of scenarios respectively weighting of ensemble members according to climate signals. Nevertheless, we appreciate the comment as it is indicating, that this is not communicated clearly and we have now given a separate and extended chapter of 'Conclusions' in a revised version of manuscript.

**Point by point response to the comments**

**Scenario approach for the seasonal forecast of Kharif flows from Upper Indus Basin**

**Muhammad Fraz Ismail and Wolfgang Bogacki**

bogacki@hs-kobelnz.de

Dear Referee No. 2

Thank you for your specific comments and suggestion, we have tried to address your concerns on the points you have mentioned below. We appreciate your precise comments and detailed suggestion for the improvement of our paper.

**COMMENT:** Section 2.2 (Line 11-14). You should provide enough quantitative evidences to explain how to divide the Upper and Lower parts within UIB.

**RESPONSE:** In section *2.3 Splitting the UIB into two sub-catchments*, we have tried to explain the reasons behind the splitting in more detail. Figure 6 as well as Table 4 also go in line with the physical reasoning we have explained that section.

Ideally the split between Upper and Lower UIB model should be done downstream of the Tibetan Plateau. Due to the lack of discharge data in this part of the UIB, Kharmong as the closest gauging station where the daily discharge data is available was selected.

**Table 4: Depletion of snow cover area for Upper and Lower UIB during March 2003**

| Elevation (m asl) | 3500 | 4000 | 4500 | 5000 | 5500 | >5500 |
|---|---|---|---|---|---|---|
| 1st March | | | | | | |
| Lower UIB | 66% | 82% | 88% | 87% | 83% | 94% |
| Upper UIB | 58% | 79% | 58% | 51% | 58% | 71% |
| 1st April | | | | | | |
| Lower UIB | 42% | 71% | 84% | 84% | 78% | 92% |
| Upper UIB | 24% | 50% | 48% | 43% | 51% | 73% |

[Figure]

**Figure 6: Snow cover variation in the month of March and April 2003 in UIB**

**COMMENT:** Section 2.2 (Line 15-16). There are little details about the Kirpich travel-time equation that is used to determine the 3-day time lag between Kharmong and Tarbela. Please add the relevant information in the revised version.

**RESPONSE:** We have now explained in the revised manuscript under section *2.3 Splitting the UIB into two sub-catchments* that how the Kirpich travel time equation was applied.

Kirpich equation (5) (Kirpich, 1940; USAD, 2010)

$$t = 0.00195 \, L^{0.77} \, S^{-0.385} \tag{5}$$

as in this empirical equation the time of concentration $t$ [min] is only related to the length of the main channel $L$ [m] and the slope of the longest hydraulic length $S$ [-]. Given the altitudes of Kharmong and Darband (upstream Tarbela reservoir) gauging stations as 2,542 and 436 m a.s.l respectively and a channel length[1] of about 617 km, the approximated time-lag of 5000 min was finally rounded to 3 days.

**COMMENT:** Section 3 (Line 1-5). This part (forecast skill metrics) belongs to method description. Please move it to Section 2.

**RESPONSE:** You are right. In our revised paper, we have now made a section *2.7 Verification methods* as you have pointed out.

**COMMENT:** The authors only focused on the forecast performance of median (50%) volume values. Actually, the extreme volumes, like "dry" (20%) and "wet" (80%) conditions, may be of greater importance for the downstream regions. It should add a few additional skill assessments in terms of predicting extreme conditions.

**RESPONSE:** We have now included Table 7 for the comparison of forecast skill with different available methods. We have made a "dry" (20%) and "wet" (80%) comparison for SRM+G for the Kharif flow volumes as shown in Figure 12 and for calculating the PSS and the RPSS. We have also prepared a Taylor diagram (see Figure 13) for the model performance.

**Table 7: Comparison of forecast skills between IRSA, UBC, and SRM+G**

| Model | MAE | RMSE | MPE | MAPE | R | AC$_u$ | PSS |
|-------|-----|------|-----|------|---|--------|-----|
|       | km$^3$ | km$^3$ | % | % | - | - | - |
| IRSA | 6.5 | 8.0 | 5.8 | 10.9 | 0.107 | 0.085 | -0.070 |
| UBC | 6.9 | 7.7 | 6.3 | 11.4 | 0.318 | 0.260 | 0.096 |
| SRM+G | 6.0 | 7.0 | -2.0 | 9.5 | 0.223 | 0.168 | -0.079 |
* * *
[1] digitised from Esri's World Imagery. Source: Esri, DigitalGlobe, GeoEye, i-cubed, USDA, USGS, AEX, Getmapping, Aerogrid, IGN, IGP, swisstopo, and the GIS User Community

[Figure]

**Figure 12: Comparison of Kharif flow forecast with 20% and 80% quantiles of SRM+G scenario ensembles years 2003 - 2016**

[Figure]

**Figure 13: Taylor diagram of IRSA, UBC, and SRM+G model performance**

**COMMENT:** This study assessed the forecast skill only by examining the volume difference of determined values. In the revised version, a few probabilistic quantitative metrics, like anomaly correlation (AC), Brier Score (BS), the false alarm rate (FAR), hit rate (HR), and Equitable Threat Score (ETS), should be employed to assess its skill in probabilistic forecasting.

**RESPONSE:** See response of the previous comment. Where we have given a table for the comparison of forecast skills.

**COMMENT:** Section 3. Please add 1-2 figures to illustrate the comparison of SRM+G model-based scenario approach with other forecast models.

**RESPONSE:** We have provided a comparison of all three operational forecast models as given in Table 6.

**Table 6: Comparison of Kharif flow volumes [km³] 2003 - 2016**

| Year | Observed | IRSA | UBC | SRM+G |
|------|----------|------|------|-------|
| 2003 | 67.8 | 64.0 | 63.5 | 63.1 |
| 2004 | 51.8 | 60.5 | 63.6 | 60.8 |
| 2005 | 68.9 | 69.0 | 73.3 | 60.9 |
| 2006 | 67.8 | 68.4 | 73.3 | 61.6 |
| 2007 | 60.5 | 74.9 | 70.1 | 61.0 |
| 2008 | 57.7 | 68.5 | 59.2 | 53.9 |
| 2009 | 57.6 | 63.7 | 67.2 | 62.4 |
| 2010 | 76.6 | 63.3 | 68.4 | 61.4 |
| 2011 | 60.0 | 67.2 | 70.8 | 59.9 |
| 2012 | 55.4 | 61.3 | 61.7 | 60.4 |
| 2013 | 65.6 | 64.9 | 58.8 | 59.8 |
| 2014 | 52.9 | 64.6 | 64.2 | 61.4 |
| 2015 | 67.2 | 63.3 | 61.3 | 58.9 |
| 2016 | 66.4 | 62.4 | 66.5 | 63.1 |

**COMMENT:** You are suggested to add a comparison between SRM+G and SRM, to highlight the superiority of SRM+G in terms of incorporating glacier component.

**RESPONSE:** We have added a comparison of SRM with and without glacier component in order to see what is the possible impact of glaciers melting component in the late summer in the revised manuscript (See Figure 8 and 9).

[Figure]

**Figure 8: Comparison of SRM+G (with glaciers) and SRM (without glaciers) for Lower UIB – 2008**

[Figure]

**Figure 9: Comparison of SRM+G (with glaciers) and SRM (without glaciers) for Upper UIB - 2008**

The impact of glacier in the Lower UIB is much more prominent than in the Upper UIB. The reason is that in the Lower UIB the glaciated area is about 10.5% of the catchment area while in the Upper UIB the glaciated area is only 1.7%.

**COMMENT:** Adding a comparison of SRM+G estimates between with and without consideration of divisions (Upper and Lower parts) should be inserted and discussed in the revision.

**RESPONSE:** A detailed discussion has been added in section *3.1 Splitting of the UIB catchment*. Though the whole UIB showed an acceptable agreement between simulated and observed flows in terms of $R^2$ and $D_v$, initial hindcast results proved to be not satisfactory especially for the Early Kharif (1st April – 10th June) season, having a mean absolute percentage error MAPE of 25.9%. The MODIS snow cover analysis showed that in the Tibetan plateau already in March the snow cover is fading away rapidly while on the other hand, in the north-western part of the catchment the same elevation zone is still widely covered with snow (Figure 6). With splitting the MAPE could be reduced to 15.8%.

**COMMENT:** Section 2.3 (Line 30). If I understand correctly, R indicates the daily runoff depth, not precipitation depth.

**RESPONSE:** Yes, you are right, precipitation depth relates to the original SRM notation. We have corrected it in the revised version see section *2.2 Model structure* line 15.

**COMMENT:** Section 2.5 (Line 33). "TRMM 3B34 product" should be "TRMM 3B34 product". Please correct it.

**RESPONSE:** Corrected in revised manuscript.

**List of relevant changes**

**LIST OF CHNAGES**

Following is the list of all relevant changes made in the revised manuscript:

**Figures:**

1. Combined Figure 1 & 3 => Now revised manuscript Figure 1
2. Updated Figure 6 and 7 => Now Figure 10 and 11 in revised manuscript
3. Added new Figures 6, 7, 8, 9, 12, 13, 14 and 15 in revised manuscript

**Tables:**

1. Added new Tables 3, 5, 6 and 7 in revised manuscript

**Chapters:**

1. Corrected the mistake in units of R in chapter 2.2 Model Structure
2. Incorporated the Kirpich equation and related description in chapter 2.3 Splitting the UIB into two sub-catchments
3. Incorporated the description about the degree-day factors linear increase during the snowmelt season in chapter 2.5 Model parameters
4. Incorporated the discussion about ESP in chapter 2.6
5. Added a new chapter 2.7 verification methods and discussed forecast verification metrics like Anomaly Correlation ACu, Peirce skill score (PSS), Ranked Probability Score (RPS) and Ranked Probability Skill Score (RPSS).
6. Updated the chapter 3 Results and Discussion by incorporation sub-sections like
   a. Splitting of the UIB,
   b. Glacier melt component,
   c. Simulation model verification including Table 5 with $R^2$ and $D_v$
   d. Evaluation of forecasting skills including Table 7 with verification metrics
7. Updated the chapter 4 Conclusion
8. Updated the relevant references

**Marked-up manuscript**

[revised manuscript text omitted]

**Commented [i4]:** New text with physical reasoning of the degree-day factors increase.
Parameters Table is also included.

**2.6 Scenario approach for forecasting**

35  In the forecasting period which starts from the 1$^{st}$ of April, the four model variables temperature, precipitation, snow covered area and glacier exposed area have to be predicted for the forthcoming 6 months of the Kharif cropping season (April – September). As the level of skill of seasonal climate forecasts for the Hindukush –

Karakoram – Western Himalaya region for such a lead time is still not sufficient, a scenario approach already successfully applied in the Upper Jhelum catchment (Bogacki and Ismail, 2016) is used.

This scenario approach has a lot in common with traditional Ensemble Streamflow Prediction ESP developed at the U.S. National Weather Service as a method for generating long-term probabilistic streamflow outlooks (Day,

5 1985). Based on the assumption that past meteorology is representative of possible future events, ESP uses historical temperature and precipitation time series as forcings for the hydrological model to produce an ensemble of streamflow traces. A probabilistic forecast is created by statistical analysis of the multiple streamflow scenarios produced (Franz et al. 2008). Initial basin conditions are usually estimated by forcing the hydrological model with observed meteorology in a "warm-up" phase up to the time of forecast (Wood and Lettenmaier, 2008).

10 The seasonal scenario approach also uses historical temperature and precipitation as forcings for the SRM+G model. In contrast to ESP however, this approach is, like the other operational forecast models for UIB, primarily focussed on a deterministic forecast of total Kharif flow volume. Besides the "most likely" (median) flow, SRM+G forecasts only give an indication of the bandwidth of expected flows by the dry (20%) and wet (80%) quantiles as limits of the "likely" range.

> **Commented [i5]:** Discussion about ESP has been included.

15 Another notable difference are the initial basin conditions. SRM and SRM+G do not use any initial conditions, like soil moisture state of snow-water equivalent in other hydrological models. Instead however, the snow-cover area and the glacier exposed area are input variables to the model. For reasons of simplicity, the glacier exposed area is treated like the meteorological variables, i.e. the historical time-series are used. The depletion of the snow-covered area during the forecast period, which is the decisive factor for each forecast, is however predicted by so-called

20 "modified depletion curves". These modified depletion curves are derived from the conventional depletion curves of each elevation zone by replacing the time scale with the cumulative daily snow-melt depth (Martinec et al., 2011). The decline of the modified depletion curves depends on the initial accumulation of snow and represents the actual snow-water equivalent. When initial snow depth is low the modified depletion curve declines faster than in years when a lot of snow has accumulated. In the end of March, when the seasonal forecast is carried out, an

25 elevation zone showing already some decline in snow covered area, and hence having also some cumulated degree-days, is chosen as "key zone". Comparing the relation of decline in snow covered area versus cumulated degree-days with a statistical analysis of the modified depletion curves of previous years, the actual amount of snow is estimated and the future depletion anticipated accordingly, while assuming similar snow conditions for all elevation zones.

30 The major difference to other hydrological models as used in the ESP is the positive effect that usually the uncertainty in the initial conditions is progressively superseded by the actual meteorological conditions. In SRM+G however, if an erroneous depletion estimate is in effect then it will persist during the whole forecast period. As all ensemble traces are based on the chosen depletion curves, the initial estimate is crucially influencing each trace of the ensemble in the same direction.

> **Commented [i6]:** Discussion about ESP has been included.

**2.7 Verification methods**

[revised manuscript text omitted]

**Commented [i12]:** Updated text with respect to the new analysis being carried out in the revised manuscript.
* * *
[8] El Niño Southern Oscillation

**Acknowledgements**

The authors thank National Engineering Services Pakistan (Pvt.) Ltd. (NESPAK), Lahore and AHT Group AG, Essen, Germany that they could be part of the project team. They are highly grateful to Indus River System Authority (IRSA) and WAPDA's Glacier Monitoring Research Centre (GMRC) and Surface Water Hydrology

5 Project (SWHP) of for sharing their forecast results as well as the daily discharge data. The authors are also thankful to the two anonymous reviewers for their valuable and constructive comments that substantially helped to improve the quality of the manuscript.

**References**

[revised manuscript text omitted]

| 10-daily Period | Elevation Zone (m a.s.l) | | | | | | |
|---|---|---|---|---|---|---|---|
| | 3000 | 3500 | 4000 | 4500 | 5000 | 5500 | >5500 |
| | 1 | 2 | 3 | 4 | 5 | 6 | 7 |
| $T_{10d}$ | 2.0 | 2.0 | 2.0 | 2.0 | 0.5 | 0.5 | 0.5 |
| 1 | 0.37 | 0.35 | 0.35 | 0.52 | 0.56 | 0.48 | 0.60 |
| 2 | 0.43 | 0.40 | 0.40 | 0.59 | 0.64 | 0.54 | 0.70 |
| 3 | 0.49 | 0.45 | 0.46 | 0.66 | 0.73 | 0.80 | 0.80 |
| 4 | 0.54 | 0.51 | 0.51 | 0.73 | 0.80 | | |
| 5 | 0.60 | 0.56 | 0.56 | 0.80 | | | |
| 6 | 0.66 | 0.61 | 0.62 | | | | |
| 7 | 0.71 | 0.66 | 0.67 | | | | |
* * *
[9] 10-Daily average temperature in ºC in each elevation zone.

**Table 3: SRM+G Model Parameters for both Upper and Lower UIB**

Commented [i24]: Table about parameters has been incorporated.

[revised manuscript text omitted]

**Commented [i29]:** New figure showing MODIS snow cover information in the lower and upper UIB on 1st March and 1st April 2003.

[Figure]

[Figure]

[Figure]

**Figure 7: Monthly distribution of flow components in UIB**

**Commented [i30]:** New figure included about the monthly distribution of flow component in UIB.

[Figure]

**Figure 8**: Comparison of SRM+G (with glaciers) and SRM (without glaciers) for Lower UIB – 2008

**Commented [i31]:** New figure added for the comparison of SRM+G (with glaciers) and SRM (without glaciers).

[Figure]

**Figure 9**: Comparison of SRM+G (with glaciers) and SRM (without glaciers) for Upper UIB - 2008

**Commented [i32]:** New figure added for the comparison of SRM+G (with glaciers) and SRM (without glaciers).

[Figure]

**Figure 10**: Results of validation of final Upper UIB flow forecast model (dashed line) compared to observed flows at Kharmong (solid line) for the year 2014.

**Commented [i33]:** updated

[Figure]

**Figure 11**: Results of validation of final Lower UIB flow forecast model (dashed line) compared to observed inflows at Tarbela (solid line) for the year 2014.

**Commented [i34]:** updated

[Figure]

**Figure 12**: Comparison of Kharif flow forecast with 20% and 80% quantiles of SRM+G scenario ensembles years 2003 - 2016

[Figure]

Commented [i35]: New figure added about the dry and wet quantiles of SRM+G.

[Figure]

**Figure 13**: Taylor diagram of IRSA, UBC, and SRM+G model performance

Commented [i36]: New figure has been added for analysing the models performance.

[Figure]

**Figure 14**: Plume diagram of ensemble member traces in 2003

Commented [i37]: New figure has been added about the ensemble member trace 2003

[Figure]

**Figure 15**: Plume diagram of ensemble member traces in 2008

Commented [i38]: New figure has been added about the ensemble member trace 2008

---

## Author Response (AR3)

**Point by point response to the comments**

**Comments by Referee #02**

The authors have carefully and clearly addressed all my concerns. However, I do have two minor suggestions to further improve the current version.

1.  In the "Introduction" or "Result and discussion", I suggest the authors add more discussions to highlight the difference between yours and other studies.

    **Response:**

[revised manuscript text omitted]

**List of relevant changes**

Following are the list of all relevant changes made in the revised manuscript:

**Figures:**

1. **Figure 7** => updated in terms of using the same scale for all three components
2. **Figure 12** => updated in terms of using the proper symbol convention as used for other figures in the paper as well.
3. **Figure 16** => completely new figure as desired by the editor and referee to address the reliability.

**Tables:**

1. No change

**Chapters:**

1. New text has been added in the **introduction** to show the difference between the current and previous studies.
2. New text about reliability diagram has been added in **chap. 2.7**.
3. Discussion of reliability diagram has been added in **chap 3.4**.
4. Related references have been added in the **references** chapter.

**Marked-up manuscript**

[revised manuscript text omitted]

**Commented [B8]:** New figure: Reliability diagram as desired in Minor revisions